# Gravitationally induced decoherence vs space-time diffusion: testing the quantum nature of gravity

Jonathan Oppenheim ®[1] ✉, Carlo Sparaciari ®[1], Barbara Šoda ®[1,2,3] &
Zachary Weller-Davies[1,3]

We consider two interacting systems when one is treated classically while the other system remains quantum. Consistent dynamics of this coupling has been shown to exist, and explored in the context of treating space-time classically. Here, we prove that any such hybrid dynamics necessarily results in decoherence of the quantum system, and a breakdown in predictability in the classical phase space. We further prove that a trade-off between the rate of this decoherence and the degree of diffusion induced in the classical system is a general feature of all classical quantum dynamics; long coherence times require strong diffusion in phase-space relative to the strength of the coupling. Applying the trade-off relation to gravity, we find a relationship between the strength of gravitationally-induced decoherence versus diffusion of the metric and its conjugate momenta. This provides an experimental signature of theories in which gravity is fundamentally classical. Bounds on decoherence rates arising from current interferometry experiments, combined with precision measurements of mass, place significant restrictions on theories where Einstein's classical theory of gravity interacts with quantum matter. We find that part of the parameter space of such theories are already squeezed out, and provide figures of merit which can be used in future mass measurements and interference experiments.

When considering the dynamics of composite quantum systems, there are many regimes where one system can be taken to be classical and the other quantum-mechanical. For example, in quantum thermo-dynamics, we often have a quantum system interacting with a large thermal reservoir that can be treated classically, whilst in atomic physics it is common to consider the behaviour of quantum atoms in the presence of classical electromagnetic fields. Things become more complicated when one considers classical-quantum (CQ) dynamics where the quantum system back-reacts on the classical system. This is particularly relevant in gravity because we would like to study the back-reaction of thermal radiation being emitted from black holes in space-time, and while the matter fields can be described by quantum

field theory, we only know how to treat space-time classically. Likewise in cosmology, vacuum fluctuations are a quantum effect that we believe seeds galaxy formation, while the expanding space-time they live on can only be treated classically. In addition to the need for an effective theory that treats space-time in the classical limit, there has long been a debate about whether one should quantise gravity[1–12].

The prevailing view has been that a quantum–classical coupling is inconsistent. Many proposals for such dynamics[13,14] are not completely positive (CP), meaning they are at best an approximation and fail outside a regime of validity[15,16]. A map Λ is completely positive (CP), iff $\mathbb{1} \otimes \Lambda$ is positive. This is the required condition used to derive the GKSL equation. If it is violated, the dynamics acting on half of an entangled

[1]Department of Physics and Astronomy, University College London, Gower Street, London WC1E 6BT, UK. [2]Department of Physics, University of Waterloo, Waterloo, ON, Canada. [3]Perimeter Institute for Theoretical Physics, Waterloo, ON, Canada. ✉e-mail: j.oppenheim@ucl.ac.uk

state, give negative probabilities. The semi-classical Einstein's equation[17–19], which replaces the quantum operator corresponding to the stress-energy tensor by its expectation value, is another attempt to treat the classical limit from an effective point of view, but it is non-linear in the state, leading to pathological behaviour if quantum fluctuations are of comparable magnitude to the stress-energy tensor[20]. This is often the precise regime we would like to understand.

However, examples of classical–quantum dynamics such as those first introduced in refs. [21,22] and studied in refs. [11,23–27] do not suffer from such problems and are consistent. More generally, the master equation shown in Eq. (4)[11], is linear in the state space, preserves the division of classical degrees of freedom and quantum ones, and is completely positive (CP), and preserves normalisation. This ensures that probabilities of measurement outcomes remain positive and always add to 1. The dynamics is related to the GKSL or Lindblad equation[28,29], which for bounded generators of the dynamics, is the most general Markovian dynamics for an open quantum system. More precisely, we consider dynamics which is autonomous, meaning the couplings in the theory do not depend on time. Likewise, Eq. (4) is the most general Markovian classical–quantum dynamics with bounded generators[11,30]. Sub-classes of this master equation, along with measurement and feedback approaches, have been discussed in the context of Newtonian models of gravity[24,25,31–35] and further developed into a spatially covariant framework so that Einstein gravity in the ADM formalism[36] emerges as a limiting case[11,37]. Dynamics which is manifestly diffeomorphism invariant has also been introduced using path integral methods[38,39].

In this work, we move away from specific realisations of CQ dynamics, in order to discuss their common features and the experimental signatures that follow from this. An early precursor to the discussion here is the insight of Diósi[22], who added classical noise and quantum decoherence to the master equation of ref. [13], and found the noise and decoherence trade-off required for the dynamics to become completely positive. Here we prove that the phenomena found in refs. [11,22,26] are generic features of all CQ dynamics; the classical–quantum interaction necessarily induces decoherence on the quantum system, and there is a generic trade-off between the rate of decoherence and the amount of diffusion in the classical phase space. The stronger the interaction between the quantum system and the classical one, the greater the trade-off. One cannot have quantum systems with long-coherence times without inducing a lot of diffusion in the classical system. One can also generalise this result to a trade-off between the rate of diffusion and the strength of more general couplings to Lindblad operators, with decoherence being a special case.

## Results

Our main result is expressed as Eqs. (26) and (23), which bounds the product of diffusion coefficients and Lindblad coupling constants in terms of the strength of the CQ-interaction. It is precisely this trade-off which allows the theories considered here, to evade the no-go arguments of Feynmann[1,2], Aharonov[3], Eppley and Hannah[4] and others[1,7–9,15,16,40–48]. The essence of arguments against quantum–classical interactions is that they would prohibit superpositions of quantum systems that source a classical field. Since different classical fields are perfectly distinguishable in principle, if the classical field is in a distinct state for each quantum state in the superposition, the classical field could always be used to determine the state of the quantum system, causing it to decohere instantly. By satisfying the trade-off, the quantum system preserves coherence because diffusion of the classical degrees of freedom means that the state of the classical field does not determine the state of the quantum system[11,25]. Equation (26) and other variants we derive, quantify the amount of diffusion required to preserve any amount of coherence. If space–time curvature is treated classically, then complete positivity of

the dynamics means its interaction with quantum fields necessarily results in unpredictability and gravitationally induced decoherence.

This trade-off between the decoherence rate and diffusion provides an experimental signature, not only of models of hybrid Newtonian dynamics such as refs. [24,33] or post-quantum theories of general relativity such as refs. [11,39] but of any theory which treats gravity as being fundamentally classical. The metric and their conjugate momenta necessarily diffuse away from what Einstein's general relativity predicts. This experimental signature squeezes classical–quantum theories of gravity from both sides: if one has shorter decoherence times for superpositions of different mass distributions, one necessarily has more diffusion of the metric and conjugate momenta. In the "Methods" subsection "Detecting gravitational diffusion" we show that the latter effect causes imprecision in measurements of mass such as those undertaken in the Cavendish experiment[49–51] or in measurements of Newton's constant "Big $G$"[52–54]. The precision at which a mass can be measured in a short time, thus provides an upper bound on the amount of gravitational diffusion, as quantified by Eq. (42). In the other direction, decoherence experiments place a lower bound on the diffusion. Our estimates suggest that experimental lower bounds on the coherence time of large molecules[55–60], combined with gravitational experiments measuring the acceleration of small masses[61–63], already place strong restrictions on theories where space–time is not quantised. In the section "Physical constraints on the classicality of gravity" we show that several realisations of CQ-gravity are already ruled out, while other realisations produce enough diffusion away from general relativity to be detectable by future table-top experiments. Although the absence of such deviations from general relativity would not be as direct a confirmation of the quantum nature of gravity, such as experiments proposed in refs. [64–74] to exhibit entanglement or coherence generated by gravitons, it would effectively rule out any sensible theory that treats space–time classically. While confirmation of gravitational diffusion would suggest that space-time is fundamentally classical. Experiments to detect or bind gravitational diffusion also provide immediate-term prospects for probing the quantumness of gravity, while entanglement-based experiments will only be feasible in the long term.

The outline of the remainder of this paper is as follows. In the subsection "Classical-quantum dynamics" we review the general form of the CQ master equation of classical–quantum systems as derived in refs. [11,30]. The *CQ-map* can be represented in a manner akin to the Kraus representation[75] for quantum maps, with conditions for it to be completely positive and trace preserving (CPTP). We can perform a short time moment expansion of the CQ-map taking states at some initial time, to states at a later time. This gives us the CQ version of the Kramers–Moyal expansion[76,77], presented in the subsection "The CQ Kramers–Moyal expansion". The physical meaning of the moments is given in the subsection "Physical interpretation of the moments". Our main result is presented in the section "A trade-off between decoherence and diffusion", where we show that there is a general trade-off between decoherence of the quantum system and diffusion in the classical system. We generalise the trade-off to the case of fields in the subsection "Trade-off in the presence of fields" and in the subsection "Physical constraints on the classicality of gravity", we apply the inequality in the gravitational setting. The positivity constraints mean that the considerations do not depend on the specifics of the theory, only that it treats gravity classically, and is time-local. This allows us to discuss some of the observational implications of this result and we comment on the relevant figures of merit required in interference and precision mass measurements in order to constrain theories of gravity, as they are not always readily available in published reports. In addition to table-top constraints, we consider those due to cosmological observations. We then conclude with a discussion of our results. The "Methods" section collects or previews a number of technical results. Since this paper appeared on the arXiv, we have found that when the

decoherence-diffusion trade-off is saturated, there are two important consequences. The first is that in the continuous class of dynamics, the quantum state remains pure, conditioned on the classical trajectory[78]. The second is that in the path integral formulation, one can show that the dynamics are completely positive from the path integral alone[39]. For a generic path integral of Feynman–Vernon form[79], one typically only knows that the dynamics are completely positive if it was derived from a CPTP master equation.

## Classical–quantum dynamics

Let us first review the general map and master equation governing classical–quantum dynamics. The classical degrees of freedom are described by a differential manifold $\mathcal{M}$ and we shall generically denote elements of the classical space by $z$. For example, we could take the classical degrees of freedom to be position and momenta in which case $\mathcal{M} = \mathbb{R}^2$ and $z = (q, p)$. The quantum degrees of freedom are described by a Hilbert space $\mathcal{H}$. Given the Hilbert space, we denote the set of positive semi-definite operators with trace at most unity as $S_{\leq 1}(\mathcal{H})$. Then the CQ object defining the state of the CQ system at a given time is a map $\varrho : \mathcal{M} \to S_{\leq 1}(\mathcal{H})$ subject to a normalisation constraint $\int_{\mathcal{M}} dz \mathrm{Tr}_{\mathcal{H}}[\varrho] = 1$. To put it differently, we associate to each classical degree of freedom an un-normalised density operator, $\varrho(z)$, such that $\mathrm{Tr}_{\mathcal{H}}[\varrho] = p(z) \geq 0$ is a normalised probability distribution over the classical degrees of freedom and $\int_{\mathcal{M}} dz \varrho(z)$ is a normalised density operator on $\mathcal{H}$. An example of such a *CQ-state* is the CQ qubit depicted as a $2 \times 2$ matrix over phase space[26]. More generally, we can define any CQ operator f(z) which lives in the fibre bundle with base space $\mathcal{M}$ and fibre $\mathcal{H}$.

Just as the Lindblad equation is the most general evolution law that maps density matrices to density matrices, we can ask what is the most general evolution law that preserves the quantum-classical state-space. Any such dynamics, if it is to preserve probabilities, must be completely positive, norm-preserving, and linear in the CQ-state. That dynamics must be linear can be seen as follows: if someone prepares a system in one of two states $\sigma_0$ or $\sigma_1$ depending on the value of a coin toss ($|0\rangle\langle 0|$ with probability $p$, $|1\rangle\langle 1|$ with probability $1-p$), then the evolution $\mathcal{L}$ of the system must satisfy $p|0\rangle\langle 0| \otimes \mathcal{L}\sigma_0 + (1 - p)|1\rangle\langle 1| \otimes \mathcal{L}\sigma_1 = \mathcal{L}(p\sigma_0 + (1 - p)\sigma_1)$ otherwise the system evolves differently depending on whether we are aware of the value of the coin toss. A violation of linearity further implies that when the system is in state $\sigma_0$ it evolves differently depending on what state the system would have been prepared in, had the coin been $|1\rangle\langle 1|$ instead of $|0\rangle\langle 0|$. This motivates our restriction to linear theories. We will also require the map to be Markovian on the combined classical–quantum system, which is equivalent to requiring that there is no hidden system that acts as a memory. This is natural if the interaction is taken to be fundamental, but is the assumption that one might want to remove if one thinks of the hybrid theory as an effective description. We thus take these as the minimal requirements that any fundamental classical–quantum theory must satisfy if it is to be consistent.

The most general CQ-dynamics, which maps CQ states onto themselves can be written in the form[11]

$$\varrho(z, t + \delta t) = \int dz' \sum_{\mu\nu} \Lambda^{\mu\nu}(z|z', \delta t) L_\mu \varrho(z', t) L_\nu^\dagger \tag{1}$$

where the $L_\mu$ is an orthogonal basis of operators and $\Lambda^{\mu\nu}(z|z', \delta t)$ is positive semi-definite for each $z, z'$. Henceforth, we will adopt the Einstein summation convention so that we can drop $\sum_{\mu\nu}$ with the understanding that equal upper and lower indices are presumed to be summed over. The normalisation of probabilities requires

$$\int dz \Lambda^{\mu\nu}(z|z', \delta t) L_\nu^\dagger L_\mu = \mathbb{I}. \tag{2}$$

The choice of basis $L_\mu$ is arbitrary, although there may be one which allows for unique trajectories[26]. Equation (1) can be viewed as a generalisation of the Kraus decomposition theorem.

In the case where the classical degrees of freedom are taken to be discrete, Poulin[25] used the diagonal form of this map to derive the most general form of Markovian master equation for bounded operators, which is the one introduced in ref. 21. When the classical degrees of freedom are taken to live in a continuous configuration space, we need to be a little more careful, since $\varrho(z)$ may only be defined in a distributional sense; for example, $\varrho(z) = \delta(z, \bar{z})\varrho(\bar{z})$. In this case (1) is completely positive if the eigenvalues of $\Lambda^{\mu\nu}(z|z', \delta t), \lambda^\mu(z|z', \delta t)$, are positive so that $\int dz dz' P_\mu(z, z') \lambda^\mu(z|z', \delta t) \geq 0$ for any vector with positive components $P_\mu(z, z')$[30]. One can derive the CQ master equation by performing a short time expansion of Eq. (1) in the case when the $L_\mu$ is bounded[11]. To do so, we first introduce an arbitrary basis of traceless Lindblad operators on the Hilbert space, $L_\mu = \{I, L_a\}$. Now, at $\delta t = 0$, we know Eq. (1) is the identity map, which tells us that $\Lambda^{00}(z|z', \delta t = 0) = \delta(z, z')$. Looking at the short-time expansion coefficients, by Taylor expanding in $\delta t \ll 1$, we can write

$$\Lambda^{\mu\nu}(z|z', \delta t) = \delta_0^\mu \delta_0^\nu \delta(z, z') + W^{\mu\nu}(z|z') \delta t + O(\delta t^2). \tag{3}$$

By substituting the short-time expansion coefficients into Eq. (1) and taking the limit $\delta t \to 0$ we can write the master equation in the form

$$\frac{\partial \varrho(z, t)}{\partial t} = \int dz' \, W^{\mu\nu}(z|z') L_\mu \varrho(z') L_\nu^\dagger - \frac{1}{2} W^{\mu\nu}(z) \{L_\nu^\dagger L_\mu, \varrho\}_+, \tag{4}$$

where $\{,\}_+$ is the anti-commutator, and preservation of normalisation under the trace and $\int dz$ defines

$$W^{\mu\nu}(z) = \int dz' W^{\mu\nu}(z'|z). \tag{5}$$

We see the CQ master equation is a natural generalisation of the Lindblad equation and classical rate equation in the case of classical–quantum coupling. We give a more precise interpretation of the different terms arising when we perform the Kramers–Moyal expansion of the master equation at the end of the section. The positivity conditions from Eq. (1) transfer to positivity conditions on the master equation via (3). We can write the positivity conditions in an illuminating form by writing the short time expansion of the transition amplitude $\Lambda^{\mu\nu}(z|z', \delta t)$, as defined by Eq. (3), in block form

$$\Lambda^{\mu\nu}(z|z', \delta t) = \begin{bmatrix} \delta(z, z') + \delta t W^{00}(z|z') & \delta t W^{0\beta}(z|z') \\ \delta t W^{\alpha 0}(z|z') & \delta t W^{\alpha\beta}(z|z') \end{bmatrix} + O(\delta t^2) \tag{6}$$

and the dynamics will be positive if and only if $\Lambda^{\mu\nu}(z|z', \delta t)$ is a positive matrix. It is possible to introduce an arbitrary set of Lindblad operators $\bar{L}_\mu$ and appropriately redefine the couplings $W^{\mu\nu}(z|z')$ in Eq. (4)[11]. For most purposes, we shall work with a set of Lindblad operators that includes the identity $L_\mu = (I, L_\alpha)$; this is sufficient since any CQ master equation is completely positive if and only if it can be brought to the form in Eq. (4), where the matrix (6) is positive.

## The CQ Kramers–Moyal expansion

In order to study the positivity conditions it is first useful to perform a moment expansion of the dynamics in a classical-quantum version of the Kramers–Moyal expansion as in ref. 11. In classical Markovian dynamics, the Kramers–Moyal expansion relates the master equation to the moments of the probability transition amplitude and proves to be useful for a multitude of reasons. Firstly, the moments are related to observable quantities; for example, the first and second moments of the probability transition amplitude characterise the amount of drift

and diffusion in the system. This is reviewed in the subsection "Physical interpretation of the moments". Secondly, the positivity conditions on the master equation transfer naturally to positivity conditions on the moments, which we can then relate to observable quantities. In the classical–quantum case, we shall perform a short time moment expansion of the transition amplitude $\Lambda^{\mu\nu}(z|z', \delta t)$ and then show that the master equation can be written in terms of these moments. We then relate the moments to observational quantities, such as the decoherence of the quantum system and the diffusion in the classical system.

We work with the form of the dynamics in Eq. (4), using an arbitrary orthogonal basis of Lindblad operators $L_\mu = \{\mathbb{I}, L_\alpha\}$. We take the classical degrees of freedom $\mathcal{M}$ to be $d$ dimensional, $z = (z_1, \ldots z_d)$, and we label the components as $z_i$, $i \in \{1, \ldots d\}$. We begin by introducing the moments of the transition amplitude $W^{\mu\nu}(z|z')$ appearing in the CQ master Eq. (3)

$$D_{n, i_1 \ldots i_n}^{\mu\nu}(z') := \frac{1}{n!} \int dz W^{\mu\nu}(z|z')(z - z')_{i_1} \ldots (z - z')_{i_n}. \quad (7)$$

The subscripts $i_j \in \{1, \ldots d\}$ label the different components of the vectors $(z - z')$. For example, in the case where $d = 2$ and the classical degrees of freedom are position and momenta of a particle, $z = (z_1, z_2) = (q, p)$, then we have $(z - z') = (z_1 - z_1', z_2 - z_2') = (q - q', p - p')$. The components are then given by $(z - z')_1 = (q - q')$ and $(z - z')_2 = (p - p')$. $M_{n, i_1 \ldots i_n}^{\mu\nu}(z', \delta t)$ is seen to be an $n$th rank tensor with $d^n$ components.

In terms of the components $D_{n, i_1 \ldots i_n}^{\mu\nu}$ the short time expansion of the transition amplitude $\Lambda^{\mu\nu}(z|z')$ is given by[30]

$$\Lambda^{\mu\nu}(z|z', \delta t) = \delta_0^\mu \delta_0^\nu \delta(z, z') + \delta t \sum_{n=0}^\infty D_{n, i_1 \ldots i_n}^{\mu\nu}(z') \left(\frac{\partial^n}{\partial z_{i_1}' \ldots \partial z_{i_n}'}\right) \delta(z, z') + O(\delta t^2), \quad (8)$$

and the master equation takes the form[11]

$$\frac{\partial \varrho(z, t)}{\partial t} = \sum_{n=1}^\infty (-1)^n \left(\frac{\partial^n}{\partial z_{i_1} \ldots \partial z_{i_n}}\right) \left(D_{n, i_1 \ldots i_n}^{00}(z)\varrho(z, t)\right)$$
$$- i[H(z), \varrho(z)] + D_0^{\alpha\beta}(z) L_\alpha \varrho(z) L_\beta^\dagger - \frac{1}{2} D_0^{\alpha\beta}(z)\{L_\beta^\dagger L_\alpha, \varrho(z)\}_+ \quad (9)$$
$$+ \sum_{\mu\nu \neq 00} \sum_{n=1}^\infty (-1)^n \left(\frac{\partial^n}{\partial z_{i_1} \ldots \partial z_{i_n}}\right) \left(D_{n, i_1 \ldots i_n}^{\mu\nu}(z) L_\mu \varrho(z, t) L_\nu^\dagger\right),$$

where we define the Hermitian operator $H(z) = \frac{i}{2}(D_0^{\mu 0} L_\mu - D_0^{0\mu} L_\mu^\dagger)$ (which is Hermitian since $D_0^{\mu 0} = D_0^{0\mu*}$). We see the first line of Eq. (9) describes purely classical dynamics, and is fully described by the moments of the identity component of the dynamics $\Lambda^{00}(z|z')$. The second line describes pure quantum Lindbladian evolution described by the zeroth moments of the components $\Lambda^{\alpha 0}(z|z'), \Lambda^{\alpha\beta}(z|z')$; specifically the (block) off diagonals, $D_0^{\alpha 0}(z)$, describe the pure Hamiltonian evolution, whilst the components $D_0^{\alpha\beta}(z)$ describe the dissipative part of the pure quantum evolution. Note that the Hamiltonian and Lindblad couplings can depend on the classical degrees of freedom so the second line describes the action of the classical system on the quantum one. The third line contains the non-trivial classical-quantum back-reaction, where changes in the distribution over phase space are induced and can be accompanied by changes in the quantum state.

**Physical interpretation of the moments**
Let us now briefly review the physical interpretation of the moments that will appear in our trade-off relation. In particular, the zeroth moment determines the rate of decoherence (and Lindbladian coupling more generally), the first moment gives the force exerted by the quantum system on the classical system, and the second moment

determines the diffusion of the classical degrees of freedom. For this discussion, we shall take the classical degrees of freedom to live in a phase space $\Gamma = (\mathcal{M}, \omega)$, where $\omega$ is the symplectic form.

Consider the expectation value of any CQ operator $O(z)$, $\langle O(z)\rangle := \int dz \text{Tr}[O(z)\varrho]$ which does not have an explicit time dependence. Its evolution law can be determined via Eq. (9)

$$\frac{d\langle O\rangle}{dt} = \int dz \text{Tr}\left[O(z)\frac{\partial\varrho}{\partial t}\right] = \int dz \text{Tr}\varrho$$
$$\left[-i[O(z), H(z)] + D_0^{\alpha\beta}(z) L_\beta^\dagger O(z) L_\alpha - \frac{1}{2} D_0^{\alpha\beta}(z)\{L_\alpha L_\beta^\dagger, O(z)\}_+ \quad (10)\right.$$
$$\left. + \sum_{n=1}^\infty D_{n, i_1 \ldots i_n}^{\mu\nu}(z) \left(\frac{\partial^n}{\partial z_{i_1} \ldots \partial z_{i_n}}\right) \left(L_\nu^\dagger L_\mu O(z, t)\right)\right],$$

where we have used cyclicity of trace and integration by parts, to bring the equation of motion into a form that would enable us to write a CQ version of the *Heisenberg representation*[11] for a CQ operator. If we are interested in the expectation value of phase space variables $O(z) = z_i \mathbb{I}$ then Eq. (10) gives

$$\frac{d\langle z_i\rangle}{dt} = \int dz D_{1, i}^{\mu\nu} \text{Tr}\left[L_\nu^\dagger L_\mu \varrho(z, t)\right] \quad (11)$$

with all higher order terms vanishing, and we see that $\sum_{\mu\nu \neq 00} D_{1, i}^{\mu\nu} \langle L_\nu^\dagger L_\mu\rangle$ governs the average rate at which the quantum system moves the classical system through phase space, and with the back-reaction is quantified by the Hermitian matrix $D_1^{\alpha\mu} := (D_1^{\text{br}})^{\alpha\mu}$. The force of this back-reaction is especially apparent if the equations of motion are Hamiltonian in the classical limit as in ref. 11. I.e. if we define $H_I(z) := h^{\alpha\beta} L_\beta^\dagger L_\alpha$ and take $D_{1, i}^{\alpha\beta} = \omega_i^j d_j h^{\alpha\beta}$ with $\omega$ the symplectic form and $d_j$ the exterior derivative. Then Eq. (11) is analogous to Hamilton's equations, and the CQ evolution equation after tracing out the quantum system has the form of a Liouville's equation to first order and in the classical limit,

$$\frac{\partial \rho(z, t)}{\partial t} = \{H_c, \rho(z, t)\} + \text{tr}(\{H_I(z), \varrho(z)\}) + \ldots \quad (12)$$

with $\rho(z) := \text{Tr}[\varrho(z)]$.

The significance of the second moment is also seen via Eq. (10) to be related to the variance of phase space variables $\sigma_{z_{i_1} z_{i_2}} := \langle z_{i_1} z_{i_2}\rangle - \langle z_{i_1}\rangle\langle z_{i_2}\rangle$

$$\frac{d\sigma_{z_{i_1}, z_{i_2}}^2}{dt} = 2\langle D_{2, i_1, i_2}^{\alpha\beta} L_\beta^\dagger L_\alpha\rangle + \langle z_2 D_{1, z_{i_1}}^{\alpha\beta} L_\beta^\dagger L_\alpha\rangle - \langle z_{i_2}\rangle\langle D_{1, z_{i_1}}^{\alpha\beta} L_\beta^\dagger L_\alpha\rangle$$
$$+ \langle z_{i_1} D_{1, z_{i_2}}^{\alpha\beta} L_\beta^\dagger L_\alpha\rangle - \langle z_{i_1}\rangle\langle D_{1, z_{i_2}}^{\alpha\beta} L_\beta^\dagger L_\alpha\rangle. \quad (13)$$

In the case when $D_{1, z_{i_1}}$ is uncorrelated with $z_{i_2}$ and $D_{1, z_{i_2}}$ uncorrelated with $z_{i_1}$, then the growth of the variance only depends on the diffusion coefficient.

The zeroth moment $D_0^{\alpha\beta}$ is just the pure Lindbladian couplings. The simplest example is the case of a pure decoherence process with a single Hermitian Lindblad operator $L$ and decoherence coupling $D_0$. Then we can define a basis $\{|a\rangle\}$ via the eigenvectors of $L$ and

$$\left\langle a \left|\frac{\partial\varrho}{\partial t}\right| b\right\rangle = -i\langle a|[H(z), \varrho]|b\rangle - \frac{1}{2} D_0(L(a) - L(b))^2 \langle a|\varrho|b\rangle, \quad (14)$$

and we see that the matrix elements of $\varrho$ which quantify coherence between the states $|a\rangle, |b\rangle$ decay exponentially fast with a decay rate of $D_0(L(a) - L(b))^2$. For a damping/pumping process of a quantum harmonic oscillator with Hamiltonian $H = \omega a^\dagger a, L_\downarrow = a, L_\uparrow = a^\dagger$, $a$ the creation operator, and $D_0^{\uparrow\uparrow}, D_0^{\downarrow\downarrow}$ the non-zero couplings, then standard

calculations[26,80] show that an initial superposition $\frac{1}{\sqrt{2}}|n+m\rangle$ with $n, m$ large and $n \gg m$ will initially decohere at a rate of approximately $(D_0^{\uparrow\uparrow} + D_0^{\downarrow\downarrow})(m+n)/2$, and the state will eventually thermalise to a temperature of $\omega/\log(D_0^{\downarrow\downarrow}/D_0^{\uparrow\uparrow})$. So in this case, the Lindblad couplings not only determine the rate of decoherence but also the rate at which energy is pumped into the harmonic oscillator. In the next section, we will derive the trade-off between Lindblad couplings and the diffusion coefficients. Although we will sometimes refer to this as a trade-off between decoherence and diffusion, this terminology is only strictly appropriate for pure decoherence processes, while more generally, it is a trade-off between Lindblad couplings and diffusion coefficients.

## A trade-off between decoherence and diffusion

In this section, we present our main result by using positivity conditions to prove the trade-off between decoherence and diffusion seen in models such as those of refs. [11,22,26] are in fact a general feature of all classical-quantum interactions. We shall also generalise this, and derive a trade-off between diffusion and arbitrary Lindbladian coupling strengths. The trade-off is in relation to the strength of the dynamics and is captured by Eqs. (20), (23) and (26). In the subsection "Trade off in the presence of fields" we extend the trade-off to the case where the classical and quantum degrees of freedom can be fields and use this to show that treating the metric as being classical necessarily results in diffusion of the gravitational field.

There are two separate possible sources for the force (or *drift*) of the back-reaction of the quantum system on phase space—it can be sourced by either the $D_{1,i}^{0\alpha}$ components or the Lindbladian components $D_{1,i}^{\alpha\beta}$. We shall deal with both sources simultaneously by considering a CQ Cauchy-Schwartz inequality which arises from the positivity of

$$\text{Tr}\left[\int dz dz' \Lambda^{\mu\nu}(z|z') O_\mu(z,z')\rho(z') O_\nu^\dagger(z,z')\right] \geq 0 \quad (15)$$

for any vector of CQ operators $O_\mu$. One can verify that this must be positive directly from the positivity conditions on $\Lambda^{\mu\nu}(z|z')$ and we go through the details in the Appendix section "Positivity conditions and the trade-off between decoherence and diffusion". A common choice for $O_\mu$ would be the set of operators $L_\mu = \{\mathbb{I}, L_\alpha\}$ appearing in the master equation.

The inequality in Eq. (15) turns out to be especially useful since it can be used to define a (pseudo) inner product on a vector of operators with components $O_\mu$ via

$$\langle \bar{O}_1, \bar{O}_2 \rangle = \int dz dz' \text{Tr}\left[\Lambda^{\mu\nu}(z|z') O_{1\mu}\varrho(z') O_{2\nu}^\dagger\right] \quad (16)$$

where $||\bar{O}|| = \sqrt{\langle \bar{O}, \bar{O} \rangle} \geq 0$ due to Eq. (15). Technically this is not positive definite, but this shall not be important for our purpose. Taking the combination $O_\mu = ||\bar{O}_2||^2 O_{1\mu} - \langle \bar{O}_1, \bar{O}_2 \rangle O_{2\mu}$ for vectors $O_{1\mu}, O_{2\mu}$, positivity of the norm gives

$$\begin{aligned} ||\bar{O}||^2 &= ||(||\bar{O}_2||^2 \bar{O}_1 - \langle \bar{O}_1, \bar{O}_2 \rangle \bar{O}_2)||^2 \\ &= ||\bar{O}_2||^2 (||\bar{O}_1||^2||\bar{O}_2||^2 - |\langle \bar{O}_1, \bar{O}_2 \rangle|^2) \geq 0, \end{aligned} \quad (17)$$

and as long as $||\bar{O}_2|| \neq 0$ we have a Cauchy–Schwartz inequality

$$||\bar{O}_1||^2 ||\bar{O}_2||^2 - |\langle \bar{O}_1, \bar{O}_2 \rangle|^2 \geq 0. \quad (18)$$

We can use Eq. (18) to get a trade-off between the observed diffusion and decoherence by picking $O_{2\mu} = \delta_\mu^\alpha L_\alpha$ and $O_{1\mu} = b^i(z-z')_i L_\mu$, where $L_\mu = \{\mathbb{I}, L_\alpha\}$ are the Lindblad operators appearing in the master equation and $b^i$ are the components of an arbitrary vector. In this case $||\bar{O}_2|| = \int dz \text{Tr}\left[D_0^{\alpha\beta} L_\alpha \varrho L_\beta^\dagger\right]$ and one can verify using CQ Pawula theorem[30] that in order to have non-trivial back-reaction on the

quantum system complete positivity demands that $||\bar{O}_2|| > 0$, meaning the Cauchy-Schwartz inequality in Equation (18) must hold. To reach this conclusion one can insert the CQ state into the CQ Cauchy–Schwartz inequality and repeat the proof of the Pawula theorem[30], which must now hold once averaged over the state. By using the short-time moment expansion of $\Lambda^{\mu\nu}(z|z')$ defined in Eq. (8) and using integration by parts, we then arrive at the observational trade-off between decoherence and diffusion

$$\int dz \text{Tr}\left[2b^{i*}D_{2,ij}^{\mu\nu}b^j L_\mu \varrho(z) L_\nu^\dagger\right] \int dz \text{Tr}\left[D_0^{\alpha\beta} L_\alpha \varrho(z) L_\beta^\dagger\right] \geq \left|\int dz \text{Tr}\left[b^i D_{1,i}^{\mu\alpha} L_\mu \varrho(z) L_\alpha^\dagger\right]\right|^2, \quad (19)$$

which must hold for any positive CQ state $\varrho(z)$. Stripping out the $b^i$ vectors, (19) is equivalent to the matrix positivity condition

$$0 \preceq 2\langle D_2 \rangle \langle D_0 \rangle - \langle D_1^{br} \rangle \langle D_1^{br} \rangle^\dagger, \quad \forall \varrho(z), \quad (20)$$

where we define

$$\begin{aligned} \langle D_0 \rangle &= \int dz \text{Tr}\left[D_0^{\alpha\beta} L_\alpha \varrho(z) L_\beta^\dagger\right], \quad \langle D_1^{br} \rangle_i = \int dz \text{Tr}\left[D_{1,i}^{\mu\alpha} L_\mu \varrho(z) L_\alpha^\dagger\right], \\ \langle D_2 \rangle_{ij} &= \int dz \text{Tr}\left[D_{2,ij}^{\mu\nu} L_\mu \varrho(z) L_\nu^\dagger\right]. \end{aligned} \quad (21)$$

Since Eq. (20) holds for all states, the tightest bound is provided by the infimum over all states

$$0 \preceq \inf_{\varrho(z)}\{2\langle D_2 \rangle \langle D_0 \rangle - \langle D_1^{br} \rangle \langle D_1^{br} \rangle^\dagger\}. \quad (22)$$

The quantities $\langle D_2 \rangle$ and $\langle D_0 \rangle$ appearing in Eq. (20) are related to observational quantities. In particular $\langle D_2 \rangle$ is the expectation value of the amount of classical diffusion which is observed and $\langle D_0 \rangle$ is related to the amount of decoherence on the quantum system. The expectation value of the back-reaction matrix $\langle D_1^{br} \rangle$ quantifies the amount of back-reaction on the classical system. In the trivial case $D_1^{br} = 0$, Eq. (20) places little restriction on the diffusion and Lindbladian rates appearing on the left-hand side. We already knew from refs. [28,29] that the $D_0^{\alpha\beta}$ must be a positive semi-definite matrix, and we also know that diffusion coefficients must be positive semi-definite. However, in the non-trivial case, the larger the back-reaction exerted by the quantum system, the stronger the trade-off between the diffusion coefficients and Lindbladian coupling. Equation (20) gives a general trade-off between observed diffusion and Lindbladian rates, but we can also find a trade-off in terms of a theory's coupling coefficients alone. We show in the Appendix section "General trade-off between decoherence and diffusion coefficients" that the general matrix trade-off

$$D_1^{br} D_0^{-1} D_1^{br\dagger} \preceq 2D_2 \quad (23)$$

holds for the matrix whose elements are the couplings $D_{2,ij}^{\mu\nu}, D_{1,i}^{\alpha\mu}, D_0^{\alpha\beta}$ for any CQ dynamics. Moreover, $(\mathbb{I} - D_0 D_0^{-1}) D_1^{br} = 0$, which tells us that $D_0$ cannot vanish if there is non-zero back-reaction. Equation (23) quantifies the required amount of decoherence and diffusion in order for the dynamics to be completely positive. In Eq. (23), and throughout, $D_0^{-1}$ is the generalised inverse of $D_0^{\alpha\beta}$, since $D_0^{\alpha\beta}$ is only required to be positive semi-definite. In the special case of a single Lindblad operator $\alpha = 1$ and classical degree of freedom, and when the only non-zero couplings are $D_0^{11} := D_0, D_{2,pp}^{00} := 2D_2$ and $D_{1,q}^0 = 1$ this trade-off reduces to the condition $D_2 D_0 \geq 1$ used in ref. [22].

As a more general example, let us consider the class of theories that are continuous in phase space, and whose back-reaction is generated by a classical-quantum Hamiltonian $\hat{H}^{(m)}$ which is only a

function of the canonical coordinates $q_i$[30]. These are given by

$$\frac{\partial \varrho}{\partial t} = \{H_c, \varrho\} - i\left[\hat{H}^{(m)}, \varrho\right] + \frac{1}{2}\left\{\hat{H}^{(m)}, \varrho\right\} - \frac{1}{2}\left\{\varrho, \hat{H}^{(m)}\right\}$$
$$+ \{p^i, \{p^j, D_{2,ij}\varrho\}\} + \frac{1}{2}D_{0,ij}\left[\frac{\partial \hat{H}^{(m)}}{\partial q_i}, \left[\varrho, \frac{\partial \hat{H}^{(m)}}{\partial q_j}\right]\right] \quad (24)$$

where $H_c$ is the purely classical Hamiltonian, $p^j$ are the conjugate momenta, and $D_0$ and $D_2$ are $q_i$ dependent matrices with elements $D_{0,ij}$ and $D_{2,ij}$. Then the trade-off (23) implies that they must obey the matrix equation $2D_2 D_0 \le 1$.

It is also useful to try to obtain an observational trade-off in terms of the total drift due to back-reaction as calculated in Eq. (11)

$$\langle D_1^T \rangle_i = \sum_{\mu\nu \neq 00} \int dz \, \mathrm{Tr}\left[D_{1,i}^{\mu\nu} L_\mu \varrho(z) L_\nu^\dagger\right]. \quad (25)$$

It follows directly from Eq. (20) that when the back-reaction is sourced by either $D_{1,i}^{0\mu}$ or $D_{1,i}^{\alpha\beta}$ we can arrive at the observational trade-off in terms of the total drift

$$0 \preceq 8\langle D_2\rangle\langle D_0\rangle - \langle D_1^T\rangle\langle D_1^T\rangle^\dagger, \quad \forall \varrho(z), \quad (26)$$

where the quantities appearing in Eq. (26) are now all observational quantities, related to drift, decoherence and diffusion as outlined in the previous subsection "Physical interpretation of the moments". We believe that Eq. (26) should hold more generally, though we don't have a general proof.

In the case where the back-reaction is Hamiltonian at first order in the sense of Eq. (12), then Eq. (26) can be written as

$$\left\langle \omega \cdot \frac{\partial H_I}{\partial \vec{z}}\right\rangle\left\langle \omega \cdot \frac{\partial H_I}{\partial \vec{z}}\right\rangle^\dagger \preceq 8\langle D_2\rangle\langle D_0\rangle, \quad \forall \varrho(z). \quad (27)$$

As a result, we can derive a trade-off between diffusion and decoherence for any theory that reproduces this classical limit and treats one of the systems classically.

To summarise, whenever the back-reaction of the quantum system on the classical system induces a force on the phase space, then we have a trade-off between the amount of diffusion on the classical system and the strength of decoherence on the quantum system (or more precisely the strength of the Lindbladian couplings $D_0^{\alpha\beta}$). This can be expressed both as a condition on the matrix of coupling coefficients in the master equation, via Eq. (23) or in terms of observable quantities using Eqs. (20) and (26). In the case when the back-reaction is Hamiltonian, we further have Equation (27). We would like to apply this trade-off to the case of gravity in the non-relativistic, Newtonian limit. In order to do so, we will need to generalise the trade-off to the case of quantum fields interacting with classical ones, which we do in the subsection "Trade-off in the presence of fields". The goal will be to understand the implications of treating the metric (or Newtonian potential) as being classical by using the trade-off when the quantum back-reaction induces a force on the gravitational field which, on expectation, is the same as the weak field limit of General Relativity.

**Trade-off in the presence of fields**

We would like to explore the trade-off in the gravitational setting and explore the consequences of treating the gravitational field as being classical and matter quantum. Since gravity is a field theory, we must first discuss classical-quantum master equations in the presence of fields. In the field-theoretic case, both the Lindblad operators and the phase space degrees of freedom can have spatial dependence, $z(x), L_\mu(x)$ and a general bounded CP map which preserves the

classicality of the two systems can be written[11]

$$\rho(z, t) = \int dz' dx dy \Lambda^{\mu\nu}(z|z'; t; x, y) L_\mu(x, z, z')\varrho(z', 0)L_\nu^\dagger(y, z, z'), \quad (28)$$

where, as is usually the case with fields, in Eq. (28) it should be implicitly understood that a smearing procedure has been implemented. We elaborate on some of the details when fields are introduced in the Appendix section "Classical-quantum dynamics with fields". The condition for Eq. (28) to be completely positive on all CQ states is that for all vectors at $x$ with components $A_\nu(y, z, z')$

$$\int dz dx dy A_\mu^*(x, z, z')\Lambda^{\mu\nu}(z|z'; x, y)A_\nu(y, z, z') \ge 0 \quad (29)$$

meaning that $\Lambda^{\mu\nu}(x, y)$ can be viewed as a positive matrix in $\mu\nu$ and a positive kernel in $x, y$. In the field-theoretic case, one can still perform a Kramers–Moyal expansion and find a trade-off between the coefficients $D_0(x, y), D_1(x, y), D_2(x, y)$ appearing in the master equation. The coefficients now have an $x, y$ dependence, due to the spatial dependence of the Lindblad operators. The coefficients $D_1(x, y), D_2(x, y)$ still have a natural interpretation as measuring the amount of force (drift) and diffusion, whilst $D_0(x, y)$ describes the purely quantum evolution on the system and can be related to decoherence.

Using the positivity condition in Eq. (29) we find the same trade of between coupling constants in Eq. (23) but where now $D_2(x, y)$ is the $(p+1)n \times (p+1)n$ matrix-kernel with elements $D_{2,ij}^{\mu\nu}(x, y), D_1^{br}(x, y)$ is the $(p+1)n \times p$ matrix-kernel with rows labelled by $\mu i$, columns labelled by $\beta$, and elements $D_{1,i}^{\mu\beta}(x, y)$, and $D_0(x, y)$ is the $p \times p$ decoherence matrix-kernel with elements $D_0^{\alpha\beta}(x, y)$. Here $i \in \{1, ..., n\} \alpha \in \{1, ..., p\}$ and $\mu \in \{1, ..., p+1\}$. In the field-theoretic trade off we are treating the objects in Eq. (23) as matrix-kernels, so that for any position-dependent vector $b_\mu^i(x), (D_2 b)_i^\mu(x) = \int dy D_{2,ij}^{\mu\nu}(x, y)b_\nu^j(y)$, whilst for any position-dependent vector $a_\beta(x), (D_0 a)^\alpha(x) = \int dy D_0^{\alpha\beta}(x, y)a_\beta(y)$. Explicitly, we find that positivity of the dynamics is equivalent to the matrix condition

$$\int dx dy [b^*(x), a^*(x)]\begin{bmatrix} 2D_2(x, y) & D_1^{br}(x, y) \\ D_1^{br}(x, y) & D_0(x, y) \end{bmatrix}\begin{bmatrix} b(y) \\ a(y) \end{bmatrix} \ge 0 \quad (30)$$

which should be positive for any position-dependent vectors $b_\mu^i(x)$ and $a_\alpha(x)$. This is equivalent to trade-off between coupling constants in Eq. (23) if we view (23) as a matrix-kernel equation.

In order for the theory to be diffeomorphism invariant, we expect $D_0(x, y)$ and $D_2(x, y)$ to approach delta functions. We will not assume this, but we shall assume that the drift back-reaction is local, so that $D_1^{br}(x, y) = \delta(x, y)D_1^{br}(x)$. As we shall see in the next section, this is a natural assumption if we want to have back-reaction which is given by a local Hamiltonian. However, one might not want to assume that the form of the Hamiltonian remains unchanged to arbitrarily small distances. With this locality assumption, Eq. (30) gives rise to the same trade-off of Eq. (23), where the trade-off is to be interpreted as a matrix kernel inequality. Writing this out explicitly we have

$$\int dx dy a_\nu^{i*}(x)D_{1,i}^{\mu\alpha}(x)(D_0^{-1})_{\alpha\beta}(x, y)D_{1,j}^{\beta\nu}(x')a_\nu^i(x') \le \int dx dy 2a_\mu^{i*}(x)D_{2,ij}^{\mu\nu}(x, y)a_\nu^j(y), \quad (31)$$

where asking that this inequality holds for all vectors $a_\mu^i(x)$ is equivalent to the matrix-kernel trade-off condition of Eq. (23).

We give two examples of master equations satisfying the coupling constant trade-off in the Appendix section "Examples of Kernels saturating the decoherence diffusion coupling constants trade-off". The decoherence-diffusion trade-off tells us how much diffusion and

stochasticity are required to maintain coherence when the quantum system back-reacts on the classical one. If the interaction between the classical and quantum degrees of freedom is dictated by unbounded operators, such as the mass density, then there can exist states for which the back-reaction can be made arbitrarily large. This is the case for a quantum particle interacting with its Newtonian potential through its mass density at arbitrarily short distances. Hence, if one considers a particle in a superposition of two peaked mass densities, then there can be an arbitrarily large response in the Newtonian potential around those points, and either there must be an arbitrary amount of diffusion, or the decoherence must occur arbitrarily fast. The former is unphysical, while the latter turns out to be the case in simple examples of theories such as those discussed in the Methods subsection "Decoherence rates".

Since our goal is to experimentally constrain classical-quantum theories of gravity, we shall hereby ask that the map (28) is CP when acting on all physical states $\rho$. If one allows for arbitrarily peaked mass distributions then the coupling constant trade-off of Eq. (31) should be satisfied. In the field-theoretic case, we can similarly find an observational trade-off, relating the expected value of the diffusion matrix $\langle D_2(x,y) \rangle$ to the expected value of the drift in a physical state $\varrho$ as we did in subsection "A trade-off between decoherence and diffusion". This is done explicitly in the "Methods" subsection "Classical-quantum dynamics with fields", using a field-theoretic version of the Cauchy–Schwartz inequality given by Eq.n (73), we find

$$2\langle D_2(x,x)\rangle \int dx' dy' \langle D_0(x',y')\rangle \succcurlyeq \langle D_1^{br}(x)\rangle\langle D_1^{br}(x)\rangle^\dagger, \qquad (32)$$

where Eq. (34) is to be understood as a matrix inequality with entries

$$\langle D_0(x,y)\rangle = \int dz \, \mathrm{Tr}\left[D_0^{\alpha\beta} L_\alpha(x)\varrho L_\beta^\dagger(y)\right], \qquad (33a)$$

$$\langle D_1^{br}(x,y)\rangle_i = \int dz \, \mathrm{Tr}\left[D_{1,i}^{\mu\alpha} L_\mu(x)\varrho L_\alpha^\dagger(x)\right], \qquad (33b)$$

$$\langle D_2(x,y)\rangle_{ij} = \int dz \, \mathrm{Tr}\left[D_{2,ij}^{\mu\nu} L_\alpha(x)\varrho L_\beta^\dagger(y)\right]. \qquad (33c)$$

Similarly, when the back-reaction is sourced by either $D_{1}^{0\mu}$ or $D_{1}^{\alpha\beta}$ it follows from Eq. (32) we can arrive at the observational trade-off in terms of the total drift due to back-reaction

$$8\langle D_2(x,x)\rangle \int dx' dy' \langle D_0(x',y')\rangle \succcurlyeq \langle D_1^T(x)\rangle\langle D_1^T(x)\rangle^\dagger, \qquad (34)$$

where

$$\langle D_1^T(x)\rangle_i = \int dz \, \mathrm{Tr}\left[D_{1,i}^{0\alpha}(x)\varrho L_\alpha^\dagger(x) + D_{1,i}^{\alpha0}(x) L_\alpha \varrho(x) + D_{1,i}^{\alpha\beta}(x) L_\alpha(x)\varrho L_\beta^\dagger(x)\right]. \qquad (35)$$

We shall now use the trade-off to study the consequences of treating the gravitational field classically. We will consider the back-reaction of the mass on the gravitational field to be governed by the Newtonian interaction (or more accurately, a weak field limit of General Relativity). We shall then find that experimental bounds on coherence lifetimes for particles in superposition require large diffusion in the gravitational field in order to be maintained and this can be upper bounded by gravitational experiments.

To summarise this section, we have derived the trade-off between decoherence and diffusion for classical–quantum field theories, both in terms of coupling constants of the theory and in terms of observational quantities. This trade-off puts tight observational constraints on classical theories of gravity which we now discuss.

## Physical constraints on the classicality of gravity

In this section, we apply the trade-off of Eq. (30) to the case of gravity. A number of classical-quantum models of Newtonian gravity have been proposed[24,31–33], but since the trade-offs derived in the previous section depend only on the back-reaction, or drift term, they are insensitive to the particulars of the theory. We shall consider the Newtonian, non-relativistic limit of a classical gravitational field which we reproduce in the "Methods" subsection "Newtonian limit of CQ theory". A fuller discussion, including a derivation of the Newtonian limit starting from the covariant theories of refs. 11,39 can be found in ref. 81. It is in taking this limit where some care should be taken, since one is gauge fixing the full general relativistic theory. We denote $\Phi$ to be the Newtonian potential and in the weak field limit of General Relativity, it has a conjugate momenta we denote by $\pi_\Phi$. We assume:

(i) The theory satisfies the assumptions used to derive the master equation as in subsections "Trade-off in the presence of fields"; in particular that the theory be a completely positive norm-preserving Markovian map, and that we can perform a short-time Kramers–Moyal expansion as in "Methods" subsection "Classical-quantum dynamics with fields".

(ii) We apply the theory to the weak field limit of General Relativity, whereas recalled in "Methods" subsection "Newtonian limit of CQ theory" the Newtonian potential interacts with matter through its mass density $m(x)$,

$$H_I(\Phi) = \int d^3x \Phi(x) m(x). \qquad (36)$$

and the conjugate momentum to $\Phi$ satisfies

$$\dot{\pi}_\Phi = \frac{\nabla^2\Phi}{4\pi G} - m(x), \qquad (37)$$

where in the $c \to \infty$ limit the momentum constraint $\pi_\Phi \approx 0$ is imposed and we recover Poisson's equation for the Newtonian potential. We assume this limit of General Relativity is satisfied on expectation, at least to leading order. This may be an overly strong assumption, since the weak field limit may cease to be valid at short distances when the diffusion becomes large. A relativistic treatment is initiated in (J. Oppenheim and A. Russo, manuscript in preparation). It is also worth noting that General Relativity has not been tested at distances shorter than the millimeter scale, and here we assume it holds to arbitrarily short distances.

(iii) In relating $D_0$ to the decoherence rate of a particle in superposition, we shall assume that the state of interest is well approximated by a state living in a Hilbert space of fixed particle number. We believe this is a mild assumption: ordinary non-relativistic quantum mechanics is described via a single particle Hilbert space, and we frequently place composite massive particles in superposition and they do not typically decay into multiple particles.

(iv) We will assume that the diffusion kernel $D_2(\Phi, x, x')$ does not depend on $\pi_\Phi$ i.e. it is *minimally coupled*. This is reasonable, since in the purely classical case matter couples to the Newtonian potential and not its conjugate momenta.

With these assumptions, and treating the matter density as a quantum operator $\hat{m}(x)$, this tells us that in order for the back-reaction

term to reproduce the Newtonian interaction on average

$$
\begin{aligned}
\mathrm{Tr}\left[\{H_I, \varrho\}\right] &= \mathrm{Tr}\left[\int \mathrm{d}^3 x\, \hat{m}(x)\frac{\delta\varrho}{\delta\pi_\Phi}\right] \\
&= -\sum_{\mu\nu\neq 00}\mathrm{Tr}\left[\int \mathrm{d}^3 x D_{1,\pi_\Phi}^{\mu\nu}(\Phi,\pi_\Phi,x)L_\mu(x)\frac{\delta\varrho}{\delta\pi_\Phi}L_\nu^\dagger(x)\right],
\end{aligned}
\tag{38}
$$

then we must pick

$$
\langle D_{1,\pi_\phi}^{\mathsf{T}}(\Phi,\pi_\Phi,x)\rangle = -\langle\hat{m}(x)\rangle,
\tag{39}
$$

meaning that the back-reaction matrix $D_{1,\pi_\Phi}^{\mu\alpha}$ is nonvanishing. In the "Methods" subsection "Newtonian limit of CQ theory" we give examples of master equations for which Eq. (39) is satisfied, but their details are irrelevant since we only require the expectation of the back-reaction force to be the expectation value of the mass—a necessary condition for the theory to reproduce Newtonian gravity.

As a consequence of the coupling constant and observational trade-offs derived in Eqs. (31) and (32), a non-zero $D_{1,\pi_\Phi}$ implies that there must be diffusion in the momenta conjugate to $\pi_\Phi$. This diffusion is equivalent to adding a stochastic random process $J(x, t)$ (the Langevin picture), to the equation of motion (37) to give

$$
\dot{\pi}_\Phi = \frac{\nabla^2\Phi}{4\pi G} - m(x) + u(\Phi,\hat{m})J(t,x),
\tag{40}
$$

where we allow some *colouring* to the noise via a function $u(\Phi,\hat{m})$ which can depend on $\Phi$, and the matter distribution $\hat{m}$ (assumption (iv)). The noise process satisfies

$$
\mathbb{E}_{m,\Phi}[uJ(x,t)]=0, \quad \mathbb{E}_{m,\Phi}[uJ(x,t)uJ(y,t')]=2\langle D_2(x,y,\Phi)\rangle\delta(t,t'), \tag{41}
$$

where we have defined $\langle D_2(x,y,\Phi)\rangle = \mathrm{Tr}[D_2^{\mu\nu}(x,y,\Phi)L_\mu(x)\rho L_\nu^\dagger(y)]$, and $\rho$ is the quantum state for the decohered mass density. Here the $m,\Phi$ subscripts of $\mathbb{E}_{m,\Phi}$ allow for the possibility that the statistics of the noise process can be dependent on the Newtonian potential and mass distribution of the particle. The restriction on $\mathbb{E}_{m,\Phi}[uJ(x,t)]$ follows from assumption (ii). If $uJ(x,t)$ is Gaussian, Eq. (41) completely determines the noise process, but in general, higher-order correlations are possible, although they need not concern us here, since we are only interested in bounding the effects due to $D_2(x,y,\Phi)$.

In the non-relativistic limit, where $c\to\infty$, we impose the momentum constraint $\pi_\Phi\approx 0$ and we recover Poisson's equation for gravity, but with a stochastic contribution to the mass. This is precisely as expected on purely physical grounds: in order to maintain coherence of any mass in superposition, there must be noise in the Newtonian potential and this must be such that we cannot tell which element of the superposition the particle will be in, meaning the Newtonian potential should look like it is being sourced in part by a random mass distribution. In other words, the trade-off requires that the stochastic component of the coupling obscures the amount of mass $m$ at the different points in space where the mass may be found.

In the case where $u$ is independent of $\Phi$, it is simple to solve Eq. (40) in terms of Green's function for Poisson's equation as in the "Methods" subsection "Detecting gravitational diffusion". A formal treatment of solutions to non-linear stochastic integrals of the more general form of Eq. (40) can be found in ref. 82. A higher precision calculation would involve a full simulation of CQ dynamics, for example using unravelling methods[26,78] or the path integral as in ref. 81. However, care should be taken, as we have found that relativistic corrections put constraints on the degree of diffusion even at low energy (J. Oppenheim and A. Russo, manuscript in preparation), and one should bear this in mind when drawing conclusions on the models presented here.

In[30] it was shown that there are two classes of CQ dynamics, at least in the sense that there are those with continuous trajectories in phase space and those which contain discrete jumps. For the class of continuous CQ models (see ref. 24 and Appendix section "Continuous master equation"), we know that $J(x, t)$ should be described by a white noise process in time, and its statistics should be independent of the mass density of the particle.

For the discrete class (see ref. 11 and J. Oppenheim, "The constraints of a continuous realisation of post-quantum-classical gravity", manuscript in preparation) and "Methods" subsection "Discrete master equation"), $J(x, t)$ can involve higher order moments, and will generally be described by a jump process[26,30]. Its statistics can also depend on the mass density, since in general the diffusion matrix $D_{2,ij}^{\mu\nu}$ couples to Lindblad operators. It is worth noting that the discrete CQ theories considered in[11,26,37] generically suppress higher order moments, and often we expect that we can approximate the dynamics by a Gaussian process, but this need not be the case in general.

The stochastic contribution to the Newtonian potential leads to observational consequences which can be used to experimentally test and constrain CQ theories of gravity for various choices of kernels appearing in the CQ master equation. One immediate consequence is that the variation in Newtonian potential leads to a variation of force experienced by a particle or composite mass via $\vec{F}_{\mathrm{tot}} = -\int \mathrm{d}^3 x m(x)\nabla\Phi(x)$. We can also estimate the time-averaged force via $\frac{1}{\Delta T}\int_0^{\Delta T}\vec{F}_{\mathrm{tot}}$ where $\Delta T$ is the time resolution over which the force is measured and is the useful quantity when compared with experiments. In the "Methods" subsection "Table-top experiments" we impose the constraint $\pi\approx 0$ in Eq. (40) and find that the variance of the magnitude of the time-averaged force experienced by a particle in a Newtonian potential is given by Eq. (146),

$$
\begin{aligned}
\sigma_{\mathrm{F}}^2 = \frac{2G^2}{\Delta T}\int \mathrm{d}^3x \mathrm{d}^3y \mathrm{d}^3x'\mathrm{d}^3y'\, m(x)m(y) \\
\frac{(\vec{x}-\vec{x}')\cdot(\vec{y}-\vec{y}')}{|x-x'|^3|y-y'|^3}\langle D_2(x',y',\Phi)\rangle,
\end{aligned}
\tag{42}
$$

where the variation is averaged over the time resolution $\Delta T$. We will use this to estimate the variation in precision measurements of mass, such as modern versions of the Cavendish experiment for various choices of $\langle D_2(x',y',\Phi)\rangle$.

On the other hand, experimentally measured decoherence rates can be related to $D_0$. The important point is that the decoherence rate is dominated by the background Newtonian potential $\Phi_b$ due to the Earth. In the "Methods" subsection "Decoherence rates", we show that for a mass whose quantum state is a superposition of two states $|L\rangle$ and $|R\rangle$ of approximately orthogonal mass densities $m_L(x), m_R(x)$, and whose separation we take to be larger than the correlation range of $D_0(x,y)$, the decoherence rate is given by

$$
\lambda = \frac{1}{2}\int \mathrm{d}x\mathrm{d}y D_0^{\alpha\beta}(x,y)(\langle L|L_\beta^\dagger(y)L_\alpha(x)|L\rangle + \langle R|L_\beta^\dagger(y)L_\alpha(x)|R\rangle).
\tag{43}
$$

Via the coupling constant trade-off, Eqs. (42) and (43) then give rise to a double-sided squeeze on the coupling $D_2$. Equation (42) upper bounds $D_2$ in terms of the uncertainty of acceleration measurements seen in gravitational torsion experiments, whilst the coupling constant trade-off Eq. (43) lower bounds $D_2$ in terms of experimentally measured decoherence rates arising from interferometry experiments.

We now show this for various choices of diffusion kernel, with the details given in the "Methods" subsection "Table-top experiments". The bounds are summarised in Table 1. The diffusion coupling strength will be characterised by the coupling constant $D_2$, which we take to be a dimension-full quantity with units $\mathrm{kg}^2\,\mathrm{sm}^{-3}$, and is related to the rate of diffusion for the conjugate momenta of the Newtonian potential. We upper bound $D_2$ by considering the variation of the time averaged

**Table 1 | Current experimental bounds on classical-quantum theories for different master equations and functional dependence on the diffusion coefficient**

| Master equation | Diffusion kernel | Experimental squeeze |
|---|---|---|
| Continuous (ultra-local, non-rel.) | $D_2(\Phi; x, y) = D_2(\Phi)\delta(x, y) D_2(\Phi) = \sum_n c^n\Phi^n$ | $10^{-41} \geq D_2 \geq 10^{-9}$ kg$^2$ sm$^{-3}$ (Eq. (44)) |
| Continuous (Eq. (116) or (118)) | $D_2(\Phi; x, y) = -l_p^2 D_2(\Phi)\nabla^2\delta(x, y) D_2(\Phi) = \sum_n c^n\Phi^n$ | $10^{-9} \geq l_P^2 D_2 \geq 10^{-35}$ kg$^2$ sm$^{-1}$ (Eq. (47)) |
| Discrete (ultra-local) | $D_2(\Phi; x, y) = \frac{l_p^3}{m_p} D_2(\Phi)\delta(x, y) D_2(\Phi) = \sum_n c^n\Phi^n$ | $10^{-1} \geq \frac{l_P^3 D_2}{m_p} \geq 10^{-25}$ kg (Eq. (46)) |

The diffusion coefficient is bounded from above by observed acceleration variations $\sigma_a^2$ seen in precision mass experiments via Eq. (42). In all cases the master equation is assumed to saturate the bound which is used to find the lower bound the amount of diffusion on the quantum system by bounding $D_0$ from coherence rates via Eq. (43). It is seen that minimally coupled continuous models which are non-relativistic and do not create spatial correlations (we call these ultra-local) and have polynomial dependence on the Newtonian potential are ruled out, while continuous models with non-local correlations, such as the Diósi–Penrose (DP) kernel of Eq. (116) or the relativity inspired kernel of Eq. (118), and ultra-local discrete models are less constrained. Here $l_P$, $m_P$ are the Planck length and Planck mass respectively, which are required in order for the dimensions of $D_2(\Phi)$ to be the same in all cases.

acceleration $\sigma_a = \frac{\sigma_F}{M}$ for a composite mass $M$ which contains $N$ atoms which we treat as spheres of constant density $\rho$ with radius $r_N$ and mass $m_N$. We lower bound $D_2$ via the coupling constant trade-off of Eq. (30) and then by considering bounds on the coherence time for composite particles with total mass $M_\lambda$ and which are made up of $N_\lambda$ constituents, each with typical length scale when in superposition $R_\lambda$ and volume $V_\lambda$.

For continuous dynamics $\langle D_2(x, y, \Phi)\rangle = D_2(x, y, \Phi)$ since the diffusion is not associated with any Lindblad operators. Let us now consider a very natural kernel, namely $D_2(x, y; \Phi) = D_2(\Phi)\delta(x, y)$ which is both translation invariant, and does not create any correlations over space-like separated regions. We call dynamics which does not create correlations over space-like separated regions ultra-local since theories that are not of this form can still be non-signalling. This is a natural kernel from the point of view of constructing theories which are diffeomorphism invariant. We also label this model as being non-relativistic since it does not include various relativistic corrections to the diffusion.

The decoherence rate for this kernel is found in the "Method" subsection "Decoherence rates" and follows immediately from Eq. (137). For a nucleon of mass $M_\lambda$ and wavepacket volume $V_\lambda$, it is $\lambda = 2D_0 M_\lambda^2/V_\lambda$. In general, the squeeze will depend on the functional choice of $D_2(\Phi)$ on the Newtonian potential. However, in the presence of a large background potential $\Phi_b$, such as that of the Earth's, we will often be able to approximate $D_2(\Phi) = D_2(\Phi_b)$. This is true for kernels that depend on $\Phi$ and $\nabla\Phi$, though the approximation does not hold for all kernels, for example $D_2 \sim -\nabla^2\Phi$ of Eq. (118) which creates diffusion only where there is mass density. For diffusion kernels $D_2(\Phi_b)$ where the background potential is dominant, we find the promised squeeze on $D_2(\Phi_b)$

$$\frac{\sigma_a^2 N r_N^4 \Delta T}{V_b G^2} \geq D_2 \geq \frac{N_\lambda M_\lambda^2}{V_\lambda \lambda}, \tag{44}$$

where $V_b$ is the volume of space over which the background Newtonian potential is significant. $V_b$ enters since the variation in acceleration is found to be

$$\sigma_a^2 \sim \frac{D_2 G^2}{r_N^4 N \Delta T} \int d^3x' D_2(\Phi_b), \tag{45}$$

where the $d^3x'$ integral is over all space.

This immediately rules out continuous theories of Newtonian gravity with noise everywhere, i.e., with a diffusion coefficient independent of the Newtonian potential, since the integral will diverge. We consider the relativistic case elsewhere.

Standard Cavendish-type classical torsion balance experiments[49] measure accelerations of the order $10^{-7}$ m s$^{-2}$ over minutes $\Delta T \sim 10^2$, so a very conservative bound is $\sigma_a \sim 10^{-7}$ m s$^{-2}$, whilst for a kg mass $N \sim 10^{26}$ and $r_N \sim 10^{-15}$ m. Conservatively taking $V_b \sim r_E^2 h$ m$^3$ where $r_E$ is the radius of the Earth and $h$ is the atmospheric height gives $D_2 \leq 10^{-41}$ kg$^2$ sm$^{-3}$. The decoherence rate $\lambda$ is bounded by various experiments[83]. Typically, the goal of such experiments is to witness interference patterns of molecules that are as massive as possible.

Taking a conservative bound on $\lambda$, for example, that arising from the interferometry experiment of[59] which saw coherence in large organic fullerene molecules with total mass $M_\lambda = 10^{-24}$ kg over a timescale of $0.1s$, gives an upper bound on the decoherence rate $\lambda < 10^1$ s$^{-1}$. Fullerene molecules are made up of $N_\lambda \sim 10^3$ particles with a typical atomic size $10^{-15}$ m. After passing through the slits the molecule becomes delocalised in the transverse direction on the order of $10^{-7}$ m before being detected. Since the interference effects are due to the superposition in the transverse $x$ direction, which is the direction of alignment of the gratings, it seems like a reasonable assumption to take the size of the wavepacket in the remaining $y, z$ direction to be the size of the fullerene, since we could imagine measuring the $y, z$ directions without effecting the coherence. We, therefore, take the volume $V_\lambda \sim 10^{-15}10^{-15}10^{-7}$ m$^3 = 10^{-37}$ m$^3$, which gives $D_2 \geq 10^{-9}$ kg$^2$ sm$^{-3}$, and suggests that classical–quantum theories of Newtonian gravity with ultra-local continuous noise are ruled out by experiment.

On the other hand, the discrete models appear less constrained due to the suppression of the noise away from the mass density. For example consider the ultra-local discrete jumping models, such as the one given in the section "Discrete master equation" which have $\langle D_2(x, y, \Phi_b)\rangle = \frac{l_p^3 D_2(\Phi_b)}{m_p} m(x)$, where $m_P = \sqrt{\frac{\hbar c}{G}}$ is the Planck mass and $l_P = \sqrt{\frac{\hbar G}{c^3}}$ is the Planck length, required to ensure $D_2$ has the units of kg$^2$ sm$^{-3}$. We find the squeeze on $D_2$

$$\frac{\sigma_a^2 N r_N^4 \Delta T}{m_N G^2} \geq \frac{l_P^3}{m_P} D_2 \geq \frac{M_\lambda}{\lambda}, \tag{46}$$

and plugging in the numbers tells us that discrete theories of classical gravity are not ruled out by experiments and we find $10^{-1}$ kg $\geq \frac{l_P^3}{m_P} D_2 \geq 10^{-25}$ kg.

We can also consider other noise kernels, with examples and some discussion is given in the section "Examples of Kernels saturating the decoherence diffusion coupling constants trade-off". A natural kernel is $D_2(x, y, \Phi_b) = -l_P^2 D(\Phi_b)\nabla^2\delta(x, y)$. The inverse Lindbladian kernel satisfying the coupling constants trade-off is to zeroth order in $\Phi(x)$, the Diósi-Penrose kernel $D_0(x, y, \Phi_b) = \frac{D_0(\Phi_b)}{|x-y|}$. We here consider higher-order terms such as those coming from the relativistic theory and in particular the diffusion kernel of Eq. (118). For this choice of dynamics, we find the squeeze for $D_2$ in terms of the variation in acceleration

$$\frac{\sigma_a^2 N r_N^3 \Delta T}{G^2} \geq l_P^2 D_2 \geq \frac{N_\lambda M_\lambda^2}{R_\lambda \lambda}. \tag{47}$$

Using the same numbers as for the ultra-local continuous model, with $R_\lambda \sim V_\lambda^{1/3} \sim 10^{-12}$ m we find that classical torsion experiments upper bound $D_2$ by $10^{-9}$ kg$^2$sm$^{-1} \geq l_P^2 D_2$, whilst interferometry experiments bound $D_2$ from below via $l_P^2 D_2 \geq 10^{-35}$ kg$^2$ sm$^{-1}$.

Equations (44), (46) and (47) show that classical theories of gravity are squeezed by experiments from both ways. We have here been extremely conservative, and we anticipate that further analysis, as well as near-term experiments, can tighten the bounds by orders of

magnitude. There are several proposals for table-top experiments to precisely measure gravity, some of which have recently been performed, and which could give rise to tighter upper bounds on $D_2$. Some of these experiments involve millimeter-sized masses whose gravitational coupling is measured via torsional pendula[61,62], or rotating attractors[63]. With such devices, the gravitational coupling between small masses can be measured while limiting the amount of other sources of noise. There are proposals for further mitigating the noise due to the environment, including inertial noise, gas particle collisions, photon scattering on the masses, and curvature fluctuations due to other sources[84–86]. Other experiments are based on interference between masses; for example, atomic interferometers allow for the measurement of the curvature of space-time over a macroscopic superposition[87–89].

We can get stronger lower bounds via improved coherence experiments. Typically, the goal of such experiments is to witness interference patterns of molecules that are as massive as possible, while here, we see that the experimental bound on CQ theories is generically obtained by maximising the coherence time for massive particles with as small wave-packet size $V_\lambda$.

Thus far we have considered local effects on particles due to diffusion. While this enables us to rule out some types of theories, the bounds are generally weak if one wants to rule out all of them. However, it may be possible to do so via cosmological considerations. In attempting to place experimental constraints on this diffusion, it is also worth considering other regimes, such as longer range effects which might be detected by gravitational wave detectors such as LIGO, or table-top interferometers[90,91]. We leave a detailed study of the effect of gravitational diffusion on cosmological scales and LIGO to future work. It suffices to mention that the effect will again depend on the form of the kernel $D_2(x, x')$. Our estimates (J. Oppenheim and Z. Weller-Davies, "Estimating space-time diffusion in interferometers", unpublished note) suggest that local effects from table-top experiments currently place a stronger bound on gravitational theories than LIGO currently does. In particular, unlike gravitational wave measurements, which are reasonably high-frequency events requiring extraordinarily high precision in the relative displacement of the arm length from its average, it is preferential to have a lower precision measurement, which occurs over a longer time period to allows for the diffusion in path length to build up, and with a smaller uncertainty in the average length of the arm itself. Furthermore, since the LIGO arm is kept in a vacuum, we do not expect strong bounds on discrete models where the diffusion is associated with an energy density.

## Discussion

A number of direct proposals to test the quantum nature of gravity are expected to come online in the next decade or two. These are based on the detection of entanglement between mesoscopic masses inside matter-wave interferometers[64–70,72,73]. For these experiments, some theoretical assumptions are needed: one requires that it is only gravitons that travel between the two masses and mediate the creation of entanglement. If this is the case, then the onset of entanglement implies that gravity is not a classical field. These can be thought of as experiments that if successful, would confirm the quantum nature of gravity (although other alternatives to quantum theory are possible[92]).

Here, we come from the other direction, by supposing that gravity is instead classical, and then exploring the consequences. Theories in which gravity is fundamentally classical were thought to have been ruled out by various no-go theorems and conceptual difficulties. However, these no-go theorems are avoided if one allows for non-deterministic coupling as in[11,21–26,30,37]. We have here proven that this feature is indeed necessary and made it quantitative by exploring the consequences of complete positivity on any dynamics that couples quantum and classical degrees of freedom. Complete positivity is required to ensure the probabilities of measurement outcomes remain positive throughout the dynamics. We have shown that any theory which preserves probabilities and treats one system classically is required to have fundamental decoherence of the quantum system, and diffusion in phase space, both of which are signatures of information loss. Using a CQ version of the Kramers–Moyal expansion, we have derived a trade-off between decoherence on the quantum system, and the system's diffusion in phase space. The trade-off is expressed in terms of the strength of the back-reaction of the quantum system on the classical one. We have derived the trade-off both in terms of coupling constants of the theory and in terms of observational quantities that can be measured experimentally.

In the case of gravity, the trade-off places a lower bound on the rate of diffusion of the gravitational degrees of freedom in terms of the decoherence rate of particles in superposition. We find that theories that treat gravity as fundamentally classical, are not ruled out by current experiments, however, we have been able to rule out a broad parameter space of Newtonian theories. This is done partly through table-top observations via Eqs. (44), (46) and (47). Given any diffusion kernel, we can compute the inaccuracy of mass measurements due to fluctuations in the gravitational field, and using the trade-off, we can derive a bound on the associated decoherence rate. This allows us to rule out broad classes of theories in terms of their diffusion kernel. For example, we are able to rule out a number of non-relativistic theories which back-react continuously in phase space.

Any theory that treats gravity classically has fairly limited freedom to evade the effects of the trade-off. There is the freedom to choose the diffusion or decoherence kernels $D_2(x, x')$ and $D_0(x, x')$, but the trade-off restricts one in terms of the other. Then, because of the results proven in[30], one can consider two classes of theory, those which are continuous realisations and whose diffusion can only depend on the gravitational degrees of freedom, and discrete theories whose diffusion can also depend directly on the matter fields. Examples of both classes of the theory are given in the "Methods" subsection "Newtonian limit of CQ theory". Finally, one could consider theories that do not reproduce the weak field limit of General Relativity to all distances, namely we could imagine that the interaction Hamiltonian of Eq. (36) does not hold to arbitrarily short distances, or arbitrarily high mass densities. This is reasonable since we do expect the Newtonian theory to break down at short distances where relativistic corrections at high energy affect the low-energy behaviour of the theory. One could also consider modifying $D_1(x, x')$ in some other way, for example, by making it slightly non-local, or by disallowing arbitrarily high mass densities, or by including an additional contribution such as the friction term discussed in the continuous master equation. All of these modifications would seem to violate Lorentz invariance in some way, and likely lead to observational consequences[93].

Here, we have only given an order of magnitude estimate of when gravitational diffusion will lead to appreciable deviations from Newtonian gravity. The most promising experiments bounding the diffusion appear to be table-top experiments which precisely measure the mass of an object. This is an area that is important from the perspective of weight standards, for example, those undertaken by NIST on the 1 kg mass standard K20 and K4[94]. The increased precision and measuring time of Kibble Balances[95] and atomic interferometers[87,88,96,97] would make such measurements an ideal testing ground, both to further constrain the diffusion kernel and to look for diffusion effects, whose dependence on the test mass is outlined in the "Methods" subsection "Detecting gravitational diffusion". Here, we have found that the resolution time $\Delta T$ over which variations of acceleration are estimated affects the strength of the bound, and it would be helpful if future experiments reported this value. Since we have found that CQ theories predict an uncertainty in mass measurements it is perhaps intriguing that different experiments to measure Newton's constant $G$ yield results whose relative uncertainty differs by as much as $5 \times 10^{-4} \, \mathrm{m^3 \, kg^{-1} \, s^{-2}}$, which is more than an order of magnitude larger

than the average reported uncertainty[52–54]. If one were to try and explain the discrepancy in $G$ measurements via gravitational diffusion, then for all the kernels we studied in Section 'Physical constraints on the classicality of gravity' we find that the variation in acceleration depends on $\frac{1}{\sqrt{N}}$ the number of nucleons in the test mass, so that masses with smaller volume should yield larger uncertainty and this would be the effect to look for in measurement discrepancies. The relatively large uncertainty in such measurements, also makes it challenging for table-top experiments to place strong upper bounds on gravitational diffusion.

Turning to the other side of the trade-off, improved decoherence times would further squeeze theories in which gravity remains classical. While a current experimental challenge is to demonstrate interference patterns using larger and larger mass particles, we here find the bounds in Eqs. (44) and (46) depend on the expectation of the particle's mass density in ways that depend on the particular kernel. Thus interference experiments with particles of high *mass density* rather than mass can be preferable. There are also kernels, for which the relevant quantity is the expectation of the mass density or $M_\lambda^2/V_\lambda$, which will depend on both the particle's mass $M_\lambda$ and volume $V_\lambda$ of the wave-packet used in the interference experiment, a quantity which is not always obtainable from many reports on such experiments. While this dependence might initially appear counter-intuitive, it follows from the fact that in order to relate the trade-off in terms of coupling constants to observational quantities, and in particular, the decoherence rate, we took expectation values of the relevant quantities to get a trade-off in terms of only averages. And indeed the decoherence rate, which is an expectation value, can easily depend on the wave-packet density, as we see from examples in the section "Decoherence rates".

Since we here show that all theories that treat gravity classically necessarily decohere the quantum system, another constraint on theories that treat gravity classically is given by constraints on fundamental decoherence. These are usually constrained by bounds on anomalous heating of the quantum system[98]. However, these constraints are not in themselves very strong, since fundamental decoherence effects can be made arbitrarily weak. In the simplified model in the "Methods" subsection "Newtonian limit of CQ theory", the strength of the decoherence depends on the strength of the gravitational field, thus, constraints due to heating[98–111] can be suppressed, either by scaling the Lindbladian coupling constants or by having strong decoherence effects more pronounced near stronger gravitational fields such as near black holes where one expects information loss to occur. The necessity for decoherence to heat the quantum system is further weakened by the fact that the dynamics are not Markovian on the quantum fields, if one integrates out the classical degrees of freedom, space-time acts as a memory. This potentially captures some of the non-Markovian features advocated in ref. 112, who recognised that Markovianity is a key assumption in attempts to rule out fundamental decoherence or information loss. Here, however, we see that there is less freedom than one might imagine. If the Lindbladian coupling constants are made small to reduce direct heating, the gravitational diffusion must be large. Thus, heating constraints which place bounds on $D_0(x,x')$ place additional constraints on $D_2(x,x')$. In[35], it was found that for the Newtonian models of ref. 33, large $D_2(x,x')$ creates secondary heating which further constrain the theory experimentally. The decoherence-diffusion trade-off implies that this is a general feature of all theories which treat gravity classically.

While the absence of diffusion could rule out theories where gravity is fundamentally classical, the presence of such deviations, at least on short time scales, might not by itself be a confirmation of the classical nature of gravity. Such effects could instead be caused by quantum theories of gravity whose classical limit is effectively described

by Oppenheim[11]. In other words, one might expect some gravitational diffusion, because, from an effective theory point of view, one is in a regime where space-time is behaving classically. There are even claims that holographic effects could cause stochasticity[113–115] in the gravitational field. However, the trade-off we have derived is a direct consequence of treating the background space-time as fundamentally classical. In a fully quantum theory of gravity, the interaction of the gravitational field with particles in a superposition of two trajectories will cause decoherence, but coherence can then be restored when the two trajectories converge. This is because the particle's position is entangled with the gravitational field (or dressed by it), and this entanglement is erased when the different paths of the superposition converge. This is what happens when electrons interact with the electromagnetic field while passing through a diffraction grating, yet still form an interference pattern at the screen. This is a non-Markovian effect—the which-path superposition decoheres almost immediately, but this is *false-decoherence*[116] so the amount of diffusion can be arbitrarily small and is unrelated to the coherence time of the superposition.

On the other hand, the trade-off we derived is a direct consequence of the positivity condition, which is a direct consequence of the Markovian assumption. In the non-Markovian theory where General Relativity is treated classically, one still expects the master equation to take the form found in[11], but without the matrix whose elements are $D_n^{\mu\nu}$ needing to be positive semi-definite at all times[117,118]. It would therefore be surprising, if a quantum theory of gravity predicted anything close to the level of diffusion predicted by the decoherence-vs-diffusion trade-off, as there would be no need for diffusion to explain the coherence of superpositions. The regime in which the classical–quantum theory can be regarded as an effective one is taken up in ref. 119, both to address the issue of false decoherence, and also to explore the regime in which the classical–quantum theory may be a useful tool to understand the back-reaction of quantum matter in space–time, such as during black-hole evaporation, and during inflation. If we instead regard the theory as describing a fundamentally classical space-time, then it follows from the decoherence-diffusion trade-off, that the diffusion is either fundamental or its source is not describable within quantum or classical mechanics (J. Oppenheim, "Post-quantum soup", unpublished note).

## Methods
### Positivity conditions and the trade-off between decoherence and diffusion
In this section, we will introduce two forms of positivity conditions used to prove the decoherence diffusion trade-off.

The first inequality we would like to introduce is

$$\int dz A_\mu^*(z,z')\Lambda^{\mu\nu}(z|z',\delta t)A_\nu(z,z') \geq 0, \qquad (48)$$

which holds for any $A_\mu(z,z')$ for which Eq. (48) is well defined: i.e., so that the distributional derivatives in Eq. (48) are well defined.

We can derive the positivity condition (48) from the positivity of $\Lambda^{\mu\nu}(z|z')$, which must be a positive semi-definite matrix in $\mu\nu$. More precisely, the eigenvalues of $\Lambda^{\mu\nu}(z|z')$, which we denote by $\lambda^\mu(z|z')$ must be positive. They must be positive in the distributional sense, since we allow for the case that $\lambda^\mu(z|z')$ is a positive distribution, for example $\lambda^0(z|z') \sim \delta(z-z')$. Hence we require

$$\int dz dz' \lambda^\mu(z|z')P(z,z') \qquad (49)$$

is positive for any positive smearing function $P(z,z')$. Since each $\lambda^\mu$ must be positive, we can also pick a different smearing function for

each $\mu$, so that

$$\int dz dz' \lambda^\mu(z|z') P_\mu(z,z') \tag{50}$$

should be positive for any vector $P_\mu(z,z')$ with all positive entries. We can then write the matrix $\Lambda^{\mu\nu}(z|z')$ in terms of its eigenvalues

$$\Lambda^{\mu\nu}(z|z') = U^{\mu\dagger}_\rho(z|z')\lambda^\rho(z|z')U^\nu_\rho(z|z'). \tag{51}$$

We can then see the positivity of Eq. (48) directly since

$$\int dz A^*_\mu(z,z')\Lambda^{\mu\nu}(z|z',\delta t)A_\nu(z,z') = \int dz(UA)^\dagger_\mu(z|z')\lambda^\mu(z|z')(UA)_\mu(z,z')$$
$$= \int dz|(UA)_\mu|^2(z,z')\lambda^\mu(z|z') \tag{52}$$

which is positive as a consequence of Eq. (50).

As a consequence of Eq. (48) being positive, we also know that

$$\text{Tr}\left[\int dz \Lambda^{\mu\nu}(z|z')O_\mu(z,z')\rho(z')O^\dagger_\nu(z,z')\right] \geq 0 \tag{53}$$

will be positive for any vector of operators (potentially phase space dependent) $O_\mu(z,z')$. This follows from the cyclicity of the trace and the fact that $\Lambda^{\mu\nu}(z|z')O_\nu^\dagger(z,z')O_\mu(z,z')$ will be a positive operator so long as Eq. (48) holds. A common choice of $O_\mu$ would be the Lindblad operator $L_\mu$ appearing in the master equation.

The inequality in Eq. (48) proves useful to derive positivity conditions on the coupling constants appearing in the master equation, whilst Eq. (53) is useful in deriving the observational trade-off for the continuous master equation as we shall now discuss.

## General trade-off between decoherence and diffusion coefficients

We can get a general trade-off between the decoherence and diffusion coefficients which appear in the master equation, arriving at a trade-off between the decoherence and diffusion coefficients in terms of the back-reaction drift coefficient $D^{\mu\alpha}_{1,i}$.

Consider Eq. (48), and choose $A_\mu = \delta^\alpha_\mu a_\alpha + b^i_\mu(z - z')_i$. By integrating parts over the phase space degrees of freedom, we find

$$2b^{i*}_\mu D^{\mu\nu}_{2,ij}b^j_\nu + b^{i*}_\mu D^{\mu\beta}_{1,i}a_\beta + a^*_\alpha D^{\alpha\mu}_{1,i}b^i_\mu + a^*_\alpha D^{\alpha\beta}_0 a_\beta \geq 0. \tag{54}$$

Taking $i \in \{1,...,n\}\alpha \in \{1,...,p\}$ and $\mu \in \{1,...,p+1\}$, we can write this as a matrix positivity condition

$$[b^*, a^*]\begin{bmatrix} 2D_2 & D^{br}_1 \\ D^{br}_1 & D_0 \end{bmatrix}\begin{bmatrix} b \\ a \end{bmatrix} \geq 0 \tag{55}$$

where $D_2$ is the $(p+1)n \times (p+1)n$ matrix with elements $D^{\mu\nu}_{2,ij}$, $D^{br}_1$ is the $(p+1)n \times p$ matrix with rows labelled by $\mu i$ and columns labelled by $\beta$ with elements $D^{\mu\beta}_{1,i}$ and $D_0$ is the $p \times p$ decoherence matrix with elements $D^{\alpha\beta}_0$. $D^{br}_{1,i}$ describes the quantum back-reacting components of the drift. Equation (55) is equivalent to the condition that the $((p+1)n+p) \times ((p+1)n+p)$ matrix

$$\begin{bmatrix} 2D_2 & D^{br}_1 \\ D^{br}_1 & D_0 \end{bmatrix} \succcurlyeq 0. \tag{56}$$

Since we know $D_2$ and $D_0$ must be positive semi-definite, we know from Schur decomposition that

$$2D_2 \succcurlyeq D^{br}_1 D^{-1}_0 D^{br\dagger}_1, \tag{57}$$

and $(\mathbb{I} - D_0 D_0 - 1)D^{br}_1 = 0$, where $D_0$ is the generalised inverse of $D_0$. Furthermore, if $D_0$ vanishes, then clearly $D^{br}_1$ must also vanish in order for (56) to be positive semi-definite.

## Classical–quantum dynamics with fields

In this section, we describe CQ dynamics in the case where the Lindblad operators and the phase-space degrees of freedom can have spatial dependence $z(x), L_\mu(x)$.

For the case of fields, operators $O(x)$ constructed out of local fields $\phi(x)$ will in general be unbounded and hence the Stinespring dilation theorem does not hold. This problem is a common one in the study of algebraic quantum field theory and we can get around it by considering the case in which operators of interest are obtained by smearing the local fields over bounded functionals $F$. For example, we can first smear the local field fields over a smearing function $f$, $\phi_f = \int dx \phi(x)f(x)$ and then consider bounded functions of $\phi_f$ such as $F(\phi_f) = e^{i\phi_f}$. In doing this we can write a CQ version of the Stinespring dilation theorem exactly and proceed along the lines of Oppenheim[11] to show that any completely positive CQ map can be written in the form

$$\rho'(z) = \int dz dx dy \Lambda^{\mu\nu}(z|z';x,y)L_\mu(x,z,z')\varrho(z')L^\dagger_\nu(y,z,z'), \tag{58}$$

where the positivity condition states

$$\int dz dx dy A^*_\mu(x,z,z')\Lambda^{\mu\nu}(z|z';x,y)A_\nu(y,z,z') \geq 0. \tag{59}$$

We shall assume that we deal with dynamics which can be written in Lindblad form, as is usually assumed in the unbounded case[120].

## CQ Kramers–Moyal expansion for fields

Just as in the section "The CQ Kramers–Moyal expansion", we can formally introduce the moments of the transition amplitude

$$M^{\mu\nu}_{n,i_1...i_n}(w_1,...w_n;x,y,\delta t) = \int Dz \Lambda^{\mu\nu}(z|z';x,y,\delta t)(z-z')_{i_1}(w_1)...(z-z')_{i_n}(w_n) \tag{60}$$

which we assume to exist; which might involve a smearing of the operators $z(x)$. Defining $L_0(x) = \delta(x)\mathbb{I}$, we can define the coefficients $D^{\mu\nu}_{n,i_1...i_n}$ implicitly via

$$M^{\mu\nu}_{n,i_1...i_n}(z',w_1,...w_n;x,y,\delta t) = \delta^\mu_0 \delta^\nu_0 + \delta t n! D^{\mu\nu}_{n,i_1...i_n}(w_1,...w_n;x,y,\delta t). \tag{61}$$

The characteristic function then takes the form

$$C^{\mu\nu}(u,z';x,y) = \int Dz e^{i\int dw u(w)\cdot(z(w)-z'(w))}\Lambda^{\mu\nu}(z|z';x,y) \tag{62}$$

and expanding out the exponential this takes the form

$$C^{\mu\nu}(u,z';x,y) = \sum_{n=0}^\infty \int dw_1...dw_n \frac{u_i(w_1)...u_{i_n}(w_n)}{n!}M^{\mu\nu}_{n,i_1...i_n}(z',w_1,...w_n;x,y,\delta t) \tag{63}$$

performing the inverse Fourier transform, allows us to write the transition amplitude in terms of functional derivatives of the delta

function

$$\Lambda^{\mu\nu}(z|z';x,y,\delta t) = \sum_{n=0}^{\infty} \int dw_1 \ldots dw_n \frac{M^{\mu\nu}_{n,i_1\ldots i_n}(z',w_1,\ldots w_n;x,y,\delta t)}{n!} \frac{\delta^n}{\delta z'_{i_1}(w_1)\ldots z'_{i_n}(w_n)} \delta(z,z')$$

(64)

and we can use this to write a CQ master equation in the form

$$
\begin{aligned}
\frac{\partial \varrho(z,\delta t)}{\partial t} =& \sum_{n=1}^{\infty} \int dw_1 \ldots dw_n (-1)^n \frac{\delta^n}{\delta z_{i_1}(w_1)\ldots z_{i_n}(w_n)} \left( D^{00}_{n,i_1\ldots i_n}(z,w_1,\ldots w_n)\varrho(z) \right) - i[H,\varrho(z)] \\
&+ \int dxdy D^{\alpha\beta}_0(z;x,y)L_\alpha(x)\varrho(z)L_\beta(y) - \frac{1}{2}D^{\alpha\beta}_0(z;x,y)\{L^\dagger_\beta(y)L_\alpha(x),\varrho\} \\
&+ \sum_{n=0}^{\infty}\sum_{\mu\nu\neq 00} \int dxdydw_1\ldots dw_n(-1)^n \\
&\frac{\delta^n}{\delta z_{i_1}(w_1)\ldots z_{i_n}(w_n)}\left(D^{\mu\nu}_{n,i_1\ldots i_n}(z,w_1,\ldots w_n;x,y)L_\mu(x)\varrho(z)L^\dagger_\nu(y)\right).
\end{aligned}
$$

(65)

Since we are interested in studying dynamics with local back-reaction, we shall hereby take $D^{\mu\nu}_1(z,w;x,y) = D^{\mu\nu}_1(x)\delta(x,y)\delta(x,w)$. By the decoherence diffusion trade-off, which we derive in the next subsection, this also means that the diffusion matrix $D^{\mu\nu}_{2,ij}(z,w_1,w_2,x,y)$ is lower bounded by the matrix $D^{\mu\alpha}_1(x)(D^{-1}_0)_{\alpha\beta}(x,y)D^{\beta\nu}_1(y)\delta(w_1,x)\delta(w_2,y)$. This can be seen more precisely, by taking Eq. (59) with $A_\mu(x) = \delta^\alpha_\mu a_\alpha(x) + \int dw b^i_\mu(x,w)(z-z')(x,w)$ and applying the same methods as in the subsection "Trade-off between diffusion and decoherence couplings in the presence of fields". Without loss of generality we thus take $D_2(z,w_1,w_2,x,y) = D_2(z,x,y)\delta(x,w_1)\delta(y,w_2)$.

### Trade-off between diffusion and decoherence couplings in the presence of fields

In the field-theoretic case, the positivity condition is given by Eq. (59) and we can find a trade-off between decoherence and diffusion by considering $A_\mu(x) = \delta^\alpha_\mu a_\alpha(x) + \int dx b^i_\mu(x)(z-z')(x)$. So that

$$
\begin{aligned}
\int dxdy 2b^{i*}_\mu(x)D^{\mu\nu}_{2,ij}(x,y)b^j_\nu(y) + b^{i*}_\mu(x)D^{\mu\mu}_{1,i}(x,y)a_\beta(y) + a^*_\alpha(x)D^{\alpha\mu}_{1,i}(x,y)b^i_\mu(y) \\
+ a^*_\alpha(x)D^{\alpha\beta}_0(x,y)a_\beta(y) \geq 0
\end{aligned}
$$

(66)

where we use the shorthand notation $D^{\mu\nu}_{2,ij}(z,x,y) := D^{\mu\nu}_{2,ij}(x,y)$ and similarly $D^{\alpha\mu}_{1,i}(z;x,y) := D^{\alpha\mu}_{1,i}(x,y)$.

Taking $i \in \{1,\ldots,n\}\alpha \in \{1,\ldots,p\}$ and $\mu \in \{1,\ldots,p+1\}$, we can write this as a matrix positivity condition

$$\int dxdy[b^*(x),a^*(x)] \begin{bmatrix} 2D_2(x,y) & D^{br}_1(x,y) \\ D^{br}_1(x,y) & D_0(x,y) \end{bmatrix} \begin{bmatrix} b(y) \\ \alpha(y) \end{bmatrix} \geq 0$$

(67)

where $D_2(x,y)$ is the $(p+1)n \times (p+1)n$ matrix-kernel with elements $D^{\mu\nu}_{2,ij}(x,y)$, $D^{br}_1(x,y)$ is the $(p+1)n \times p$ matrix-kernel with rows labelled by $\mu i$ and columns labelled by $\beta$ with elements $D^{\mu\beta}_{1,i}(x,y)$ and $D_0(x,y)$ is the $p \times p$ decoherence matrix-kernel with elements $D^{\alpha\beta}_0(x,y)$. $D^{br}_{1,i}$ describes the quantum back-reacting components of the drift.

Equation (67) is equivalent to the condition that the $((p+1)n+p) \times ((p+1)n+p)$ matrix of operators

$$\begin{bmatrix} 2D_2 & D^{br}_1 \\ D^{br}_1 & D_0 \end{bmatrix} \succcurlyeq 0$$

(68)

be positive semi-definite. Here we are viewing the objects of Eq. (68) as matrix-kernels, so that for any position-dependent vector $b^i_\mu(x)$, $(D_2 b)^\mu_i(x) = \int dy D^{\mu\nu}_{2,ij}(x,y)b^j_\nu(y)$.

Since we know $D_2$ and $D_0$ must be positive semi-definite, we know from Schur decomposition that

$$2D_2 \succcurlyeq D^{br}_1 D^{-1}_0 D^{br\dagger}_1$$

(69)

and

$$(\mathbb{I} - D_0 D^{-1}_0)D^{br}_1 = 0,$$

(70)

where $D^{-1}_0$ is the generalised inverse of $D_0$. Furthermore, from Equation (70), we see if $D_0$ vanishes, then clearly $D^{br}_1$ must also vanish in order for (68) to be positive semi-definite.

### Observational trade-off in the presence of fields

We can use the same methods to arrive at an observational trade-off using the field-theoretic version of the Cauchy–Schwartz inequality in Eq. (18). This arises from the positivity of

$$\text{Tr}\left[\int dzdz'dxdy \Lambda^{\mu\nu}(z|z',x,y)O_\mu(z,z',x)\rho(z')O^\dagger_\nu(z,z',y)\right] \geq 0$$

(71)

for any local vector of CQ operators $O_\mu(z,z',x)$. We have to be careful, since (71) is not in general well defined since $O_\mu$ may not be trace-class. We hence assume that we consider states $\rho(z)$ and operators $O_\mu(z,z',x)$ for which Eq. (71) is well defined. Since we are interested in getting an observational trade-off we expect this to always be the case for physical classical–quantum states $\rho(z)$.

We shall use Eq. (71) to arrive at a (pseudo) inner product on a vector of operators $O_\mu$ via

$$\langle \bar{O}_1, \bar{O}_2 \rangle = \int dzdz'dxdy \text{Tr}\left[\Lambda^{\mu\nu}(z|z'x,y)O_{1\mu}(x)\varrho(z')O^\dagger_{2\nu}(y)\right]$$

(72)

where $||\bar{O}|| = \sqrt{\langle \bar{O}, \bar{O}\rangle} \geq 0$ due to Eq. (71). Technically this is not positive definite, but again, this will not worry us. Hence, so long as $||\bar{O}_2|| \neq 0$, which holds due to the CQ inequality derived in the derivation of the Pawula theorem[30], we again have a Cauchy–Schwartz inequality

$$||\bar{O}_1||^2 ||\bar{O}_2||^2 - |\langle \bar{O}_1, \bar{O}_2\rangle|^2 \geq 0.$$

(73)

Choosing $O_{1,\mu}(x) = \delta^\alpha_\mu L_\alpha(x)$ and $O_{2,\mu}(x) = \int dx' b^i(x)(z-z')_i(x)L_\mu(x)$, one finds

$$||\bar{O}_1||^2 = \int dzdxdy \text{Tr}\left[D^{\alpha\beta}_0(z;x,y)L_\alpha(x)\varrho(z)L^\dagger_\beta(y)\right] := \langle D_0 \rangle$$

(74a)

$$||\bar{O}_2||^2 = 2\int dzdxdy \text{Tr}\left[b^{i*}(x)D^{\mu\nu}_{2,ij}(z;x,y)L_\mu(x)\varrho(z)L^\dagger_\nu(y)b^j(y)\right]$$

(74b)

$$|\langle \bar{O}_1, \bar{O}_2\rangle|^2 = \left|\int dzdx\text{Tr}\left[b^{i*}(x)D^{\alpha\nu}_{1,i}(z;x)L_\alpha(x)\varrho(z)L^\dagger_\nu(x)\right]\right|^2 := \left|\left\langle \int dx b^{i*}(x)D^{br}_{1,i}(x)\right\rangle\right|^2$$

(74c)

Taking the limit $b^i(x) \to \delta(x,\bar{x})b^i(\bar{x})$, we arrive at a local trade-off between diffusion, drift and total decoherence. In particular, using (74), the definitions of the expectation values of couplings defined in Eq. (33) and the fact that for back-reaction the expectation value of $D_0$ cannot vanish, we arrive at the observational trade-off of Eq. (34)

$$b^i(\bar{x})\left[2\langle D_{2,ij}(\bar{x},\bar{x})\rangle\langle D_0\rangle - |\langle D^{br}_{1,i}(\bar{x})\rangle|^2\right]b^j(\bar{x}) \geq 0$$

(75)

which we write in matrix form as

$$2\langle D_2(\bar{x},\bar{x})\rangle\langle D_0\rangle \succcurlyeq \langle D_1^{\text{br}}(\bar{x})\rangle\langle D_1^{\text{br}}(\bar{x})\rangle^\dagger. \tag{76}$$

It then follows directly from Eq. (76) that when the back-reaction is sourced by either $D_{1,i}^{0\mu}$ or $D_{1,i}^{\alpha\beta}$ components we can arrive at the observational trade-off in terms of the total drift

$$8\langle D_2(\bar{x},\bar{x})\rangle\langle D_0\rangle \succcurlyeq \langle D_1^{\text{T}}(\bar{x})\rangle\langle D_1^{\text{T}}(\bar{x})\rangle^\dagger, \tag{77}$$

where in Eq. (77) recall that the definition of $\langle D_1^{\text{T}}(\bar{x})\rangle^\dagger$ is given by Eq. (35) in the main body.

## A spatially averaged observational trade-off

It is also useful to note that one can arrive at a spatially averaged observational trade-off which can be used to bound all of the elements of the diffusion matrix, not just its diagonals. Specifically, taking Eq. (74) with $b^i(x) = b^i$ a constant, we arrive at the trade-off

$$8\int dxdy\langle D_2(x,y)\rangle\langle D_0\rangle \succcurlyeq \left\langle \int dx D_1^{\text{T}}(x)\right\rangle\left\langle \int dx D_1^{\text{T}}(x)\right\rangle^\dagger, \tag{78}$$

where we define the expectation matrix

$$\langle D_2(x,y)\rangle_{ij} = \int dz \text{Tr}\left[D_{2,ij}^{\mu\nu}(z;x,y)L_\mu(x)\varrho(z)L_\nu^\dagger(y)\right]. \tag{79}$$

For the Newtonian limit discussed in the main body this bounds the diffusion in terms of the total mass of the particle

$$\int dxdy\langle D_2(x,y)\rangle \geq \frac{M^2}{16\lambda}. \tag{80}$$

We can also arrive at a trade-off in terms of the effective Newtonian potential sourced by the masses by taking $b^i(x) = \frac{1}{|\bar{x}-x|}$. In this case, we find the trade-off

$$8\int dxdy\frac{\langle D_2(x,y)\rangle}{|\bar{x}-x||\bar{x}-y|}\langle D_0\rangle \succcurlyeq \left\langle \int dx \frac{D_1^{\text{T}}(x)}{|\bar{x}-x|}\right\rangle\left\langle \int dx \frac{D_1^{\text{T}}(x)}{|\bar{x}-x|}\right\rangle^\dagger \tag{81}$$

which for the Newtonian limit gives a trade-off between the diffusion matrix and the effective Newtonian potential of the particle as sourced by its expectation value

$$\int dxdy\frac{\langle D_{2,\pi_\Phi\pi_\Phi}(x,y)\rangle}{|\bar{x}-x||\bar{x}-y|} \geq \frac{\left|\int dx\frac{\langle \hat{m}(x)\rangle}{|\bar{x}-x|}\right|^2}{16\lambda} = \frac{|\langle\hat{\Phi}(\bar{x})\rangle|^2}{16G^2\lambda}, \tag{82}$$

where we have defined the effective Newtonian potential as $\langle\hat{\Phi}\rangle = -G\int dx\frac{\langle\hat{m}(x)\rangle}{|\bar{x}-x|}$.

## Newtonian limit of CQ theory

In this section we motivate the Newtonian limit of gravity used in Section 'Physical constraints on the classicality of gravity'[121]. A fuller treatment can be found in[81]. We begin with classical General Relativity in the ADM formulation[36]. To derive the Hamiltonian, we start from the 3+1 split of the four metric

$$ds^2 = -(Nc\,dt)^2 + g_{ij}\left(dx^i + N^i c\,dt\right)\left(dx^j + N^j c\,dt\right), \tag{83}$$

in which case, denoting $\phi_{\text{m}}, \pi_{\text{m}}$ as canonical variables for the matter degrees of freedom, we can write the action for minimally

coupled matter

$$S = \int d^4x\left(\pi^{ij}\frac{\partial g_{ij}}{\partial t} + \pi_{\text{m}}\frac{\partial\phi_{\text{m}}}{\partial t} - N\mathcal{H} - N^i\mathcal{H}_i\right), \tag{84}$$

where we are ignoring the boundary contributions to the action. Here,

$$\mathcal{H} \equiv \left[-\frac{c^4}{16\pi G}g^{1/2}R + \frac{16\pi G}{c^2}\frac{1}{g^{1/2}}\left(g_{ik}g_{jl}\pi^{ij}\pi^{kl} - \frac{1}{2}\pi^2\right)\right] + \mathcal{H}^{(\text{m})}, \tag{85}$$

$$\mathcal{H}_i \equiv \frac{c^3}{8\pi G}g_{ij}\nabla_k\pi^{jk} + \mathcal{H}_i^{(\text{m})}, \tag{86}$$

are the Hamiltonian and momentum constraints and $\pi^{ij}$ is defined in terms of the extrinsic curvature tensor of constant $t$ surfaces, $K_{ij}$, via

$$\pi_{ij} \equiv -\frac{c^3}{16\pi G}g^{1/2}\left(K_{ij} - Kg_{ij}\right). \tag{87}$$

It is useful to note that the matter densities $\mathcal{H}^{(\text{m})}, \mathcal{H}_i^{(\text{m})}$ can be related to the matter stress-energy $T^{\mu\nu}$ via

$$\mathcal{H}^{(\text{m})} = \sqrt{g}N^2 T^{00}, \tag{88a}$$

$$\mathcal{H}_i^{(\text{m})} = \sqrt{g}N T_i^0. \tag{88b}$$

We here take the Newtonian limit of the gravitational field to be given by

$$N = \left(1+\frac{\Phi}{c^2}\right), N^i = 0, \quad g_{ij} = \left(1-\frac{2\Phi}{c^2}\right)\delta_{ij}, \pi^{ij} = -\frac{c^2}{6}\pi_\Phi\delta^{ij}, \tag{89}$$

with $\Phi(x)$ corresponding to the Newtonian potential. The choice of $\pi^{ij} = -\frac{c^2}{6}\pi_\Phi\delta^{ij}$, is to ensure that $\pi_\Phi$ is canonically conjugate to $\Phi$. A more detailed derivation starts from the full weak field metric, and gauge fixing of the shift can be found in[81]. Here, we find the effective action can be written

$$S = \int d^4x\left(\pi_\Phi\frac{\partial\Phi}{\partial t} + \pi_{\text{m}}\frac{\partial\phi_{\text{m}}}{\partial t} - H_{\text{Newt}}\right), \tag{90}$$

where the Newtonian Hamiltonian is given by

$$H_{\text{Newt}} = H_{\text{c}} + H_0^{(\text{m})} + H_{\text{I}} \tag{91}$$

with

$$H_{\text{c}} = \int d^3x\left(-\frac{2G\pi c^2}{3}\pi_\Phi^2 + \frac{(\nabla\Phi)^2}{8\pi G}\right) \tag{92}$$

the pure gravity Hamiltonian, and

$$H_{\text{I}} = \int d^3x\Phi(x)m(x) \tag{93}$$

is the interaction Hamiltonian, from which we see that non-relativistic matter couples to the Newtonian potential through its mass density $m(x)$. In the case where we have the state of matter being described by a point particle $\delta(x-x(t))$ of mass $m$ the pure matter Hamiltonian would be

$$H_0^{(\text{m})} = mc^2 + \frac{\delta^{ij}p_i p_j}{2m}. \tag{94}$$

The equations of motion for the gravitational degrees of freedom reads

$$\dot{\Phi} = -\frac{4\pi G c^2}{3}\pi_\Phi, \tag{95}$$

$$\dot{\pi}_\Phi = \frac{\nabla^2 \Phi}{4\pi G} - m(x), \tag{96}$$

which, for $\pi_\Phi = 0$ yields the Newtonian solution for a stationary mass density. In a Louivile formulation the dynamics for the density $\rho(\Phi, \pi_\Phi, x^i, p_i)$ is given by

$$\frac{\partial \rho}{\partial t} = \{H_c + H_0^{(m)}, \rho\} - \partial_i \Phi(x)\frac{\partial \rho}{\partial p_i} + \int d^3 x\, m(x)\frac{\delta \rho}{\delta \pi_\Phi(x)}, \tag{97}$$

where the Hamiltonian and momentum constraints tell us that $\rho(\Phi, \pi_\Phi, x^i, p_i)$ should only have support over phase space degrees of freedom which satisfy the constraint $\pi_\Phi \rho(\Phi, \pi_\Phi) = 0$. From Eq. (97) we can identify the classical drift associated to the back-reaction of the matter on the gravitational field from the $m(x)\frac{\delta \rho}{\delta \pi_\Phi(x)}$ term, so that

$$D_{1,\pi_\Phi}^{br}(x) = -m(x). \tag{98}$$

In the classical-quantum case, we promote $m(x)$ to an operator $\hat{m}$. In this case Eq. (93) is the interaction Hamiltonian used in ref. 24 to study CQ gravity. We see from Eq. (97) that in any theory whose first moment reproduces the Newtonian back-reaction on average

$$\mathrm{Tr}\left[\{H_I, \varrho\}\right] = \int d^3 x\, \mathrm{Tr}\left[\hat{m}(x)\frac{\delta \rho}{\delta \pi_\Phi(x)}\right] \tag{99}$$

must have a $D_{1,\pi_\Phi}^{br}$ given by

$$\langle D_{1\pi_\Phi}^{br}(x)\rangle = -\langle \hat{m}(x)\rangle, \tag{100}$$

from which the discussion at the beginning of the section "Physical constraints on the classicality of gravity" follows.

Note that the present discussion is insensitive to the details of the theory provided it satisfies Poisson's equation on average. Nonetheless, it's interesting that when starting from the relativistic theories of refs. 11,39, we find that the weak field limit resembles the models of Tilloy and Di´osi[33], which is discontinuous in $\Phi$, rather than the continuous model of Di´osi[24]. This is because[24] allows for non-zero conjugate momentum $\pi_\Phi$ with the kinetic energy of a different sign, while in ref. 81, the momentum is set to zero via the constraint equations and gauge fixing of the lapse. The discontinuity then arises because we are operating in the $c \to \infty$ limit, while in ref. 33, the discontinuity arises due to sourcing the Newtonian potential via a weak measurement process. We refer the reader to ref. 81 for details.

## Weak field CQ master equations

Although the trade-off we derive does not depend on the particulars of the classical–quantum theory (provided it reproduces Newtonian gravity in the classical limit), we give two concrete examples for completeness. In ref. 30 we show that there are two classes of classical-quantum dynamics, one which is continuous in phase space, and one which has discrete jumps in phase space. We will give examples of each. Although they are the weak field limit of Oppenheim[11], it is worth stressing that taking the Newtonian limit entails certain coordinate choices and restrictions on the metric. For example, here, we have restricted ourselves to metrics of the form of Eq. (89). Any gauge fixing of General Relativity which is done before deriving the master equation, is

generally not equivalent to taking the master equations of Oppenheim[11], and then taking the appropriate limit[81].

## Continuous master equation

For the class of master equations with continuous back-reaction, specifying that the first moment on average satisfies Eq. (100) is enough (up to drift terms which vanish under trace) to ensure the master Equation includes a term

$$\begin{aligned}\frac{\partial \varrho}{\partial t} \approx\ & \{H_c(\Phi), \varrho\} - i[H_0^{(m)}, \varrho] + \int d^3 x\left[\hat{m}(x)\frac{\delta \varrho}{\delta \pi_\Phi} + \frac{\delta \varrho}{\delta \pi_\Phi}\hat{m}(x)\right] \\ & + \int d^3 x d^3 y \frac{\delta^2}{\delta \pi_\Phi(x)\delta \pi_\Phi(x')}(D_2(\Phi, \pi_\Phi; x, y)\varrho) \\ & + \frac{1}{2}\int d^3 x d^y D_0(\Phi, x, x')([\hat{m}(x), [\varrho, \hat{m}(y)]]),\end{aligned} \tag{101}$$

where $H_c$ is the purely classical gravity Hamiltonian. We have taken the dynamics, i.e., the drift to be local in $x$, while we allow for the decoherence and diffusion terms to have some range. In this case, the evolution law is still local but correlations can be created[122]. One can also add extra diffusion and decoherence into Eq. (101) which we do not consider here since it only leads to worse experimental bounds. However, adding additional diffusion in $\Phi$ will generally be required in order to impose the constraint $\pi_\Phi \approx 0$[81]. This master equation is close to the one considered in ref. 24, where the decoherence and diffusion kernels are chosen to be the ones discussed in the second example of Examples of Kernels saturating the decoherence diffusion coupling constants trade-off. This is the weak field limit of the simplest realisation in ref. 11. The case where the diffusion is spatially uncorrelated $D_2(x, y) = \epsilon(x - x')$ a regulator which approaches a scalar delta function corresponds to the Newtonian limit of the diffusion term $\epsilon(x - x')\{N(x)\sqrt{g(x)}, \{\sqrt{g(x')}, \varrho\}\}$. Another natural diffusion kernel is $D_2(x, y) = -D_2(1 + \Phi(x'))\Delta_{x'}\delta(x, y)$, which can be understood as the Newtonian limit of the spatial diffeomorphism invariant kernel discussed in the section "Diffeomorphism invariant kernel".

One can supplement the kernel by some mechanism to control the diffusion. For example, a friction term such as

$$\mathcal{F}(\varrho) = D_f \frac{1}{2}\int dx dx' dy \{N(x)\sqrt{g(x)}, \{\sqrt{g(x')}\epsilon(x - x'), \mathcal{H}(y)\}\varrho\}. \tag{102}$$

In the weak field limit, this would add a term proportional to

$$\mathcal{F}(\varrho) \approx \int dx dx' \frac{\delta}{\delta \pi_\Phi(x)}(\pi_\Phi(x')\varrho) \tag{103}$$

to the master equation of Eq. (101). Such a term would break Lorentz invariance since it sets a temperature scale, although this is not necessarily a deal breaker, since it is believed by many that quantum gravity is also likely to also have an anomaly. However, the friction term is a modification to $D_1(x)$, and if too large, could run afoul of precision tests of General Relativity, such as the orbital decay of binary pulsars.

## Discrete master equation

An example of a discrete master equation satisfying Eq. (100) is

$$\begin{aligned}\frac{\partial \varrho}{\partial t} \approx\ & \{H_c(\Phi), \varrho\} - i[H_0^{(m)}, \varrho] + \frac{c^2}{\hbar \tau}\int d^3 x \\ & \left[e^{\frac{\hbar \tau}{c^2}\int dy \epsilon(x - y)\left(1 + \frac{2\Phi(y)}{c^2}\right)\frac{\delta}{\delta \pi_\Phi(y)}}\left(1 - \frac{2\Phi(x)}{c^2}\right)\psi(x)\varrho\psi^\dagger(x) - \frac{1}{2}\{m(x), \varrho\}_+\right],\end{aligned} \tag{104}$$

with $\tau$ a dimensionless constant, and $\hat{m}(x) = \psi^\dagger \psi$. We have here included $\hbar$ and $c$ to make it easier to compare with experiments. To

leading order, we could drop terms proportional to $\Phi(x)/c^2$ in both the exponential and in $N\sqrt{g} \approx 1 - 2\Phi/c^2$ inside the integral over $x$. This gives

$$\frac{\partial \varrho}{\partial t} \approx \{H_c(\Phi), \varrho\} - i[H_0^{(m)}, \varrho] + \frac{c^2}{\hbar\tau}\int d^3x \left[ e^{\frac{\hbar\tau}{c^2}\int dy \epsilon(x-y)\frac{\delta}{\delta\pi_\Phi(y)}}\psi(x)\varrho\psi^\dagger(x) - \frac{1}{2}\{m(x), \varrho\}_+ \right]. \quad (105)$$

These dynamical equations are supplemented with modified constraint equations as outlined in ref. 37. In any case, the trade-off in Eq. (31) is a statement independent of constraints and constraint preservation, at least in the weak field limit.

### Examples of Kernels saturating the decoherence diffusion coupling constants trade-off

In this section, we give examples of kernels satisfying the decoherence diffusion coupling constant trade-off in Eq. (23). For any choice of kernel, we can compute the degree of diffusion it induces in precision mass measurements (see the section "Detecting gravitational diffusion") and decoherence experiments (see the section "Decoherence rates") which allows us to rule out certain kernels experimentally. Diffeomorphism invariance may single out having both $D_0(x,y)$ and $D_2(x,y)$ approach delta functions at short distances[39], but other alternatives may be possible.

As a first example, we shall take the Lindbladian coupling to be Gaussian, taking

$$D_0^{\alpha\beta}(x,y) = \frac{\lambda^{\alpha\beta}r_0^3}{m_0^2}g_\mathcal{N}(x,y) \quad (106)$$

where $g_\mathcal{N}(x,y)$ is a normalised Gaussian distribution. The mass $m_0$ is a reference mass, and we shall take it equal to the mass of the nucleons which were considered in the section "Physical constraints on the classicality of gravity", meanwhile $\lambda^{\alpha\beta}$ is a coupling constant that determines the strength of the Lindbladian.

It should be noted that with this choice of smearing function, the pure Lindbladian evolution appearing in Eq. (101) can be taken to resemble the Lindbladian part of spontaneous collapse models[60,99,123–125], except here, there is no need to think about any ad-hoc field, nor think of the collapse as being a physical process. Rather, one necessarily gets decoherence of the wave function for free, via gravitationally induced decoherence[11,25,33,35,126]

We now find the diffusion kernel $D_2(x,y)$ using the coupling constants trade-off in (23). For simplicity, we shall assume the trade-off is saturated, and we will take the back-reaction to be local, so that $(D_1^{br})_i^{\mu\alpha}(x,y) = (D_1^{br})_i^{\mu\alpha}(x)\delta(x,y)$. In this case we find

$$D_{2,ij}^{\mu\nu}(x,y) = \frac{1}{2}(D_1^{br})_i^{\mu\alpha}(x)\frac{m_0^2}{r_0^3\lambda}g_\mathcal{N}^{-1}(x,y)(D_1^{br*})_i^{\mu\alpha}(y), \quad (107)$$

where $g_\mathcal{N}^{-1}(x,y)$ is the kernel inverse of a normalised Gaussian distribution.

It is shown in ref. 127, that the inverse distribution takes the form

$$g_\mathcal{N}^{-1}(x,y) = F(x,y)g_\mathcal{N}(x,y), \quad (108)$$

where

$$F(x,y) = \prod_{i=1}^d \sum_{n=0}^N c_n(r_0)H_{2n}\left(\frac{x_i - y_i}{r_0}\right), \quad (109)$$

and the limit $N \to \infty$ is taken. In Eq. (109) $c_n(r_0) = \frac{(-1)^n(r_0)^{2n}n!}{2^n}$ and $d$ is the spatial dimension, so that $x = (x_1, x_2, ..., x_d)$.

In total then, we arrive at the expression for the $D_2$ which saturates the bound

$$D_2(x,y) = \frac{1}{2}(D_1^{br})_i^{\mu\alpha}(x)\frac{m_0^2}{r_0^3\lambda}F(x,y)g_\mathcal{N}(x,y)(D_1^{br*})_i^{\mu\alpha}(y). \quad (110)$$

If we further take the back-reaction that of the Newtonian limit in the section "Newtonian limit of CQ theory" $(D_1^{br})_i^{\mu\alpha}(x,y) = \frac{1}{2}\delta^{0m}\delta_i^{\pi_\Phi}\delta(x,y)$ then we find the $D_2$ which saturates the bound is

$$D_2(x,y) = \frac{1}{8}\frac{m_0^2}{r_0^3\lambda}F(x,y)g_\mathcal{N}(x,y). \quad (111)$$

Another example of a Lindbladian coupling which is familiar in the literature is,

$$D_0^{\alpha\beta}(x,y) = \frac{D_0^{\alpha\beta}}{|x-y|}. \quad (112)$$

For a single Lindblad operator, this is the coupling introduced in[24] used to reproduce a CQ master equation of gravity with a decoherence rate given by the Diósi–Penrose formula[128–130]. Here we consider the special case where the $x,y$ dependence of $D_0(x,y)$ is the same for all $\alpha, \beta$ which need not hold in general. The fact that it gives the same decoherence rate as Diósi–Penrose can be seen by plugging Eq. (112) into the classical-quantum master equation in Eq. (101).

To invert the kernel in Eq. (112) we use the fact that

$$-\frac{1}{4\pi}\nabla_x^2\left(\frac{1}{|x-y|}\right) = \delta(x,y), \quad (113)$$

from which one can immediately read of the generalised inverse $(D_0^{-1})_{\alpha\beta}(x,y)$ to be

$$(D_0^{-1})_{\alpha\beta}(x,y) = \frac{(D_0^{-1})_{\alpha\beta}}{4\pi}\nabla_y^2(\delta(x,y)), \quad (114)$$

where $(D_0^{-1})_{\alpha\beta}$ are the matrix elements of the generalised inverse of $D_0$. As a consequence, we find for this specific choice of kernel that the diffusion matrix saturating the coupling constants bound in Eq. (31) is

$$D_{2,ij}^{\mu\nu}(x,y) = \frac{1}{2}D_{1,i}^{\mu\alpha}(x)\frac{(D_0^{-1})_{\alpha\beta}}{4\pi}\nabla_y^2(\delta(x,y))D_{1,j}^{\beta\nu}(y), \quad (115)$$

where we have also assumed the back-reaction is local. Taking the back-reaction to further be that of the Newtonian limit of Eq. (101) $(D_1^{br})_i^{\mu\alpha}(x) = \frac{1}{2}\delta^{0m}\delta_i^{\pi_\Phi}$ we find

$$D_2(x,y) = \frac{1}{8}\frac{(D_0^{-1})}{4\pi}\nabla_y^2(\delta(x,y)). \quad (116)$$

This diffusion kernel is argued for on the grounds of having the fluctuations satisfy a Poisson equation, in ref. 24.

### Diffeomorphism invariant kernel

Attempts to derive the constraint algebra of a generally covariant CQ theory[37] (and J. Oppenheim, "The constraints of a continuous realisation of post-quantum-classical gravity", manuscript in preparation), motivates the spatial diffeomorphism invariant kernel

$$D_2^{ijkl}(x,x') = -\frac{1}{8}D\sqrt{g(x)}N(x)g^{ij}g^{kl}\Delta_{x'}\delta(x,x'), \quad (117)$$

where $\Delta_x$ is the Laplace–Beltrami operator. One can also consider the full 3 + 1 kernel, via $\Delta^{(4)}\delta(x,x')\delta(t,t')$ along with the associated Green's function of $\Delta^{(4)}$ but this is irrelevant for the Newtonian limit. It is however useful in removing the apparent asymmetry in the expressions below, since one must recall that the $\delta(x,x')$ is a scalar in the first coordinate and a tensor density in the second, and likewise $\delta(t,t')$ has an implicit lapse $N(x')$ in the second position. This kernel's weak field limit is

$$D_2^{ijkl}(x,x') = -\tfrac{1}{8}\,D\delta^{ij}\delta^{kl}(1+\Phi(x))\Delta_x\delta(x,x'), \qquad (118)$$

which is close to that of Eq. (116), but with a correction term that turns out to be important.

Using $D_1(x,x') = -\tfrac{1}{2}N\sqrt{g}\,\delta(x,x')$, the Lindbladian kernel in dimension $d$ which saturates the trade-off for this diffusion kernel is

$$D_{0,ijkl}(x,x') = \frac{1}{2d^2D}\,\sqrt{g(x)}\,N(x')g_{ij}(x)g_{jk}(x')G(x,x'), \qquad (119)$$

with $G(x,x')$ the Green's function for $-\Delta$. It is a density in the $x'$ coordinate and a scalar in $x$. In the weak field limit, and to 0th order in $\Phi(x)$, this gives the Diosi–Penrose kernel, Eq. (112).

One could also consider the kernel

$$D_2^{ijkl}(x,x') = -\frac{1}{8}\,D\sqrt{g(x)}\,g^{ij}g^{kl}\Delta_x N(x)\delta(x,x'), \qquad (120)$$

which in the weak field limit is

$$D_2^{ijkl}(x,x') = -\tfrac{1}{8}\,D\delta^{ij}\delta^{kl}\Delta\Phi(x)\delta(x,x'). \qquad (121)$$

## A comment on divergences

The kernel examples given above give rise to divergent variance in the classical degrees of freedom, since in both cases the diffusion coefficient diverges when evaluated at the same point $D_2(x,x)$. Though we do not have a general proof, this seems to be a general feature of the coupling constant trade-off: for the examples where we can compute the kernel inverse, at least one of $D_2(x,x)$ and $D_0(x,x)$ diverge. A divergent $D_2(x,x)$ generally leads to a formally divergent classical energy production, whilst a divergent Lindbladian coupling $D_0(x,x)$ can lead to a divergent energy production in the matter degrees of freedom. The latter is related to the BPS problem[98] of anomalous heating, although it isn't necessarily equivalent since some kernels may diverge and be well-behaved from the point of view of energy production. This is not an issue from a conceptual point of view, since the only reason we expect energy to be conserved is due to Noether's theorem, and Noether's theorem doesn't apply when the evolution isn't unitary.

In the standard BPS problem, energy production in open quantum field theory can be made small by renormalizing the Lindbladian coefficient $D_0(x,y)$ appearing in the master equation. Thus the problem is merely one akin to the hierarchy problem, where we are required to introduce another energy scale. However, in the case of classical–quantum coupling, the coupling constant trade-off tells us that we cannot re-normalise $D_0(x,y)$ without affecting $D_2(x,y)$. In particular, tuning the diagonals $D_0(x,x)$ to be arbitrarily small (large) has the effect of tuning $D_2(x,x)$ to be arbitrarily (large) small: heuristically, one trades energy production in the classical system with energy production in the quantum system, and the relationship is fixed by the trade-off. On expectation, the total energy could be preserved, and the back-reaction can even slow down the flow of energy, but it's unclear if this is enough.

However, it is worth noting that while $D_2(x,x')$ may appear to diverge at a single point as $x \to x'$, when integrated over test functions, $\int dx dx' D_2(x,x')f(x)f(x')$ is usually well-behaved. The kernels discussed above have this property. When it comes to physically relevant quantities, such as measuring the gravitational diffusion in tabletop experiments, it is the smeared well-behaved quantity that is physically relevant. However, in cosmology, we typically take the constraint equation of General Relativity to be exactly satisfied at each point, and so one might imagine that $\pi_\Phi^2(x)$, and hence $D_2(x,x)$ is the relevant quantity (see the discussion in the section "Detecting gravitational diffusion"). However, one can set $\pi_\Phi \approx 0$ via a gauge freedom[81]. In GR, the counterpart to the $\pi_\Phi^2$ kinetic term is $G_{ijkl}\pi^{ij}\pi^{kl}(x)$, and it's perhaps worth noting that this quantity is not positive definite. It is also important to note that here, we have taken the weak field and $c \to \infty$ limit of General Relativity. At short distances when the diffusion becomes large, we expect this approximation to break down. One possible method of studying this problem rigorously would be through the regularisation properties of the classical-quantum path integral which we introduce in[38,39].

## Decoherence rates

In this section, we relate decoherence rates to $D_0$, and also to the average $\langle D_0 \rangle = \int dz \mathrm{Tr}[D_0^{\alpha\beta}(z;x,y)L_\alpha(x)\varrho L_\beta^\dagger(y)]$. In particular, we shall show that the decoherence rate of a mass in superposition, is given by Eq. (131) in terms of the Lindblad operators and $D_0^{\alpha\beta}$, and can be related to the quantity $\langle D_0 \rangle$.

We consider the case of a quantum mass initially in a partially decohered superposition of state $|L\rangle$ and $|R\rangle$. We describe the quantum state using creation and annihilation operators $\psi(x), \psi^\dagger(x)$ on a Fock space, related to the usual momentum-based Fock operators as $\psi(x) = \int dp\, e^{i\vec{p}\cdot\vec{x}}a_{\vec{p}}$. The mass density operator is defined via $\hat{m}(x) = m\psi^\dagger(x)\psi(x)$, where $m$ is the mass of the particle. We assume that the state remains well approximated by a state with fixed particle number, and the superposition can be taken to be distributions centred around $x = x_L$ and $x = x_R$ with total mass $M$, i.e., for a one-particle state we could take $|L/R\rangle = \int d^3x f_{L/R}(x)\psi^\dagger(x)|0\rangle$. We will take them to be well separated so that $f_L(x)f_R(x) \approx 0$, and we take the separation distance to be larger than the scale of the non-locality in $D_0(x,y)$. Mathematically this means that $\langle L|D_0^{\alpha\beta}(z;x,y)L_\beta^\dagger(y)L_\alpha(x)|R\rangle \approx 0$ for any local operators $L_\alpha(x)$ and $L_\beta(y)$.

With this orthogonality condition, we can then (at least initially) consider the joint quantum classical state restricted to the two-dimensional Hilbert space of these two states so that the total quantum-classical system can be written as

$$\varrho(\Phi,\pi_\Phi,t) = \begin{pmatrix} u_L(\Phi,\pi_\Phi,t) & \alpha(\Phi,\pi_\Phi,t) \\ \alpha^*(\Phi,\pi_\Phi,t) & u_R(\Phi,\pi_\Phi,t) \end{pmatrix}, \qquad (122)$$

where $u_L(\Phi,\pi_\Phi,t)$ and $u_R(\Phi,\pi_\Phi,t)$ correspond to some subnormalised probability distribution over the classical states of the gravitational field.

We define the total quantum state $\rho_Q$ by integrating over the classical degrees of freedom

$$\rho_Q = \int D\Phi D\pi_\Phi \varrho(\Phi,\pi_\Phi,t), \qquad (123)$$

and we shall relate $\langle D_0 \rangle$ appearing in the trade-off to the decoherence rate of the off diagonals of $\rho_Q$. Integrating over the classical phase space in Eq. (9), one finds the following expression

for the evolution of $\rho_Q$

$$\frac{\partial \rho_Q}{\partial t} = \int D\phi D\pi_\Phi - i[H(\Phi, \pi_\Phi), \varrho(\Phi, \pi_\Phi)]$$
$$+ \int D\phi D\pi_\Phi \int \mathrm{dx dy} \Big[ D_0^{\alpha\beta}(\Phi, \pi_\Phi; x, y) L_\alpha(x) \varrho(\Phi, \pi_\Phi, t) L_\beta^\dagger(y)$$
$$- \frac{1}{2} D_0^{\alpha\beta}(\Phi, \pi_\Phi; x, y) \{L_\beta^\dagger(y) L_\alpha(x), \varrho(\Phi, \pi_\Phi, t)\} \Big].$$

(124)

In particular, one finds that the off-diagonals $\langle L | \frac{\partial \rho_Q}{\partial t} | R \rangle$ evolve in part according to the commutator, and in part due to the Lindbladian term

$$\int D\phi D\pi_\Phi \int \mathrm{dx dy} \Big[ \langle L | D_0^{\alpha\beta}(\Phi, \pi_\Phi; x, y) L_\alpha(x) \varrho(\Phi, \pi_\Phi, t) L_\beta^\dagger(y) | R \rangle$$
$$- \frac{1}{2} D_0^{\alpha\beta}(\Phi, \pi_\Phi; x, y) \langle L | \{L_\beta^\dagger(y) L_\alpha(x), \varrho(\Phi, \pi_\Phi, t)\} | R \rangle \Big].$$

(125)

Care must be taken however, because both the quantum Hamiltonian and the Lindbladian coupling constants depend on the classical degrees of freedom which are affected by the quantum degrees of freedom, and thus the evolution of the quantum system is non-Markovian in general.

We shall now study the two terms appearing in Eq. (125) separately, starting with the first term. Since we assume that the state is well approximated by a state with fixed particle number then the contributions to the first term in Eq. (125) only come from terms where $L_\alpha(x)$ and $L_\beta(y)$ have the same number of creation and annihilation operators. To compute the expression, one commutes through the creation operators to act on the $\langle L |$ bra, and picks up a term $f_L(x)$. Similarly, one commutes the annihilation operators to the act on the $| R \rangle$ ket, and picks up a term $f_R(y)$. As a consequence

$$\langle L | D_0^{\alpha\beta}(\Phi, \pi_\Phi; x, y) L_\alpha(x) \varrho(\Phi, \pi_\Phi, t) L_\beta^\dagger(y) | R \rangle \sim D_0^{\alpha\beta}(\Phi, \pi_\Phi; x, y) f_L(x) f_R(y) \approx 0,$$

(126)

where the last equality follows from the fact that we are taking the masses to be well separated and the range of $D_0(x, y)$ is assumed to be much less than the separation between the masses.

Hence, the evolution of the off-diagonals comes from the (off-diagonals) of the unitary evolution and the second term in Eq. (125), the so-called no-event term. The off-diagonals of the no-event term is

$$-\frac{1}{2} \int D\phi D\pi_\Phi \int \mathrm{dx dy} D_0^{\alpha\beta}(\Phi, \pi_\Phi; x, y) \langle L | \{L_\beta^\dagger(y) L_\alpha(x), \varrho(\Phi, \pi_\Phi, t)\} | R \rangle,$$

(127)

which is negative definite and acts to exponentially suppress the coherence. To see this, note that expanding out $\varrho(\Phi, \pi_\Phi, t)$ in terms of the approximate 2 dimensional Hilbert space

$$\varrho(\Phi, \pi_\Phi, t) = u_L(\Phi, \pi_\Phi, t) | L \rangle \langle L | + u_R(\Phi, \pi_\Phi, t) | R \rangle \langle R |$$
$$+ \alpha(\Phi, \pi_\Phi, t) | L \rangle \langle R | + \alpha^*(\Phi, \pi_\Phi, t) | R \rangle \langle L |,$$

(128)

and using the fact that the range of $D_0(x, y)$ is much less than the separation between the left and right masses, we can write the off-diagonals of the no-event term as

$$-\frac{1}{2} \int D\Phi D\pi D_0^{\alpha\beta}(\Phi, \pi_\Phi; x, y) \Big( \langle L | L_\beta^\dagger(y) L_\alpha(x) | L \rangle + \langle R | L_\beta^\dagger(y) L_\alpha(x) | R \rangle \Big) \langle L | \varrho(\Phi, \pi_\Phi) | R \rangle.$$

(129)

Equation (129) already expresses the fact that the off-diagonal terms will decay, and the particle will decohere at a rate determined by the integrand of Eq. (129).

We can go slightly further when in the presence of a background Newtonian potential which is dominant, such as the Earth's $\Phi_b$. The Earth's background potential dominates over small fluctuations in $\Phi$ due to the particles and we can approximate Eq. (129) by

$$-\frac{1}{2} D_0^{\alpha\beta}(x, y) (\langle L | L_\beta^\dagger(y) L_\beta^\dagger(y) L_\alpha(x) | L \rangle + \langle R | L_\beta^\dagger(y) L_\beta^\dagger(y) L_\alpha(x) | R \rangle) \langle L | \rho_Q | R \rangle,$$

(130)

where the coupling $D_0^{\alpha\beta}(x, y)$ depends on the background Newtonian potential, but is otherwise phase-space independent. The result is to exponentially decrease the coherence $\langle L | \rho_Q | R \rangle$ with a rate $\lambda$ determined by

$$\lambda = \frac{1}{2} \int \mathrm{dx dy} D_0^{\alpha\beta}(x, y) (\langle L | L_\beta^\dagger(y) L_\alpha(x) | L \rangle + \langle R | L_\beta^\dagger(y) L_\alpha(x) | R \rangle).$$

(131)

Let us now show that the $\langle D_0 \rangle$ term appearing in the trade-off (34) is always less than (twice) this decoherence rate when in the presence of a background potential. Specifically, we show that

$$\langle D_0 \rangle = \int D\Phi D\pi_\Phi \int \mathrm{dx dy} \mathrm{Tr} \Big[ D_0^{\alpha\beta}(\Phi, \pi_\Phi; x, y) L_\beta^\dagger(y) L_\alpha(x) \varrho(\Phi, \pi_\Phi) \Big] \le 2\lambda,$$

(132)

where we assume that we are in the presence of a background potential. To see this, we first expand out the CQ state in terms of Eq. (128) and use the fact that $D_0$ has a range less than the separation of the masses. We then arrive at the following expression for the left-hand side of Eq. (132)

$$\int D\Phi D\pi_\Phi \int \mathrm{dx dy} D_0^{\alpha\beta}(\Phi, \pi_\Phi; x, y) (\langle L | L_\beta^\dagger(y) L_\alpha(x) | L \rangle u_L(\Phi, \pi_\Phi, t) + \langle R | L_\beta^\dagger(y) L_\alpha(x) | R \rangle u_R(\Phi, \pi_\Phi, t)),$$

(133)

In the presence of a background potential, this dominates the contribution to the decoherence and we are left with

$$\int \mathrm{dx dy} D_0^{\alpha\beta}(x, y) (\langle L | L_\beta^\dagger(y) L_\alpha(x) | L \rangle \langle L | \rho_Q | L \rangle + \langle R | L_\beta^\dagger(y) L_\alpha(x) | R \rangle \langle R | \rho_Q | R \rangle).$$

(134)

Due to the positivity of the CQ density matrix $\langle L | \rho_Q | L \rangle$ and $\langle R | \rho_Q | R \rangle$ must both be positive. Furthermore, they must sum to one due to normalisation, from which Eq. (131) directly follows.

It is also important to note that though $\lambda$ is the decoherence rate of a particle in superposition of $L/R$ states, the bound (132) holds even for fully decohered masses in any mixture of $| L \rangle \langle L |, | R \rangle \langle R |$ states. This can be seen directly from (133) which depends only on $u_L, u_R$.

## Decoherence rate examples

In this section, we give an explicit example of a decoherence rate calculation. Importantly, we see that in general, the decoherence rate can depend on the probability density. This suggests that the terms appearing in the trade-off relation will need to depend on expectation values such as the expectation value of the mass at a point $x$, rather than a stronger bound in terms of the mass density. This is perhaps not surprising, since the decoherence rate itself can be thought of as an expectation value, being related to the average time it takes for off-diagonal elements to decay. In conclusion, this motivates us to

advocate for the volume of the wave packet to be included in the figure of merit in future interference experiments.

We take the Newtonian limit master equation defined by Eq. (101). We ignore the unitary part of the evolution, since it will not directly contribute to the decoherence rate, and can be small for a free particle in superposition. From Eq. (101) we find the relevant evolution for the quantum state $\rho_Q$, obtained by integrating over the classical degrees of freedom to be

$$\frac{\partial \rho_Q}{\partial t} = \frac{1}{2} \int d^3x d^3y D_0(x, y) \left( [\hat{m}(x), [\rho_Q, \hat{m}(y)]] \right). \quad (135)$$

We now compute the off-diagonal elements for a particle in superposition of orthogonal $|L\rangle, |R\rangle$ states

$$\left\langle L \left| \frac{\partial \rho_Q}{\partial t} \right| R \right\rangle = - \int d^3x d^3y D_0(x, y)(m_L(x) - m_R(x))(m_L(y) - m_R(y))\langle L|\rho_Q|R\rangle. \quad (136)$$

where $m_L(x) \approx \langle L|\hat{m}(x)|L\rangle$ and similarly for the right state. We see that the off-diagonals decay exponentially with a rate determined by

$$\lambda = \int d^3x d^3y D_0(x, y)(m_L(x) - m_R(x))(m_L(y) - m_R(y)). \quad (137)$$

In the main body, and the previous subsection, we have assumed that the superposition of the particle is much less than the typical scale of $D_0(x, y)$. In this example, this means that we take the particles sufficiently separated so that we can approximate $D_0(x, y)m_L(x)m_R(y) \approx 0$, in which case Eq. (137) is precisely the decoherence rate calculated in Eq. (131) with $L(x) = \hat{m}(x)$, as is to be expected.

The natural decoherence kernel from the point of view of diffeomorphism invariance is $D_0(x, y) = D_0\delta(x, y)$, in which case for a particle of mass $M$, and uniform wave-packet volume $V$, Eq. (137) gives $\lambda = 2D_0M^2/V$.

As another example, we can take $D_0(x, x')$ to be the Diosi–Penrose decoherence kernel defined via $D_0(x, y) = \frac{D_0}{|x-y|}$, so that the off-diagonals decay exponentially with a rate proportional to the Diosi–Penrose decoherence rate

$$\lambda = \int d^3x d^3y \frac{D_0}{|x - y|}(m_L(x) - m_R(x))(m_L(y) - m_R(y)). \quad (138)$$

In this example, taking the superposition to be sufficiently separated means that we are approximating $\frac{D_0}{|x_L - x_R|} \approx 0$ in comparison with the rest of the terms appearing in Eq. (138). We are then left with

$$\lambda = \int d^3x d^3y \frac{D_0}{|x - y|}(m_L(x)m_L(y) + m_R(x)m_R(y)), \quad (139)$$

which for spherical distributions of radius $R$ and total mass $M$ is proportional to the average gravitational self-energy of each mass distribution $\lambda = \frac{6D_0M^2}{5R}$.

For a composite particle of mass $M$, made up of $N$ constituents each of radius $R$, the mass density will be represented by a sum over all of the particles $m(x) = \sum_i m_i(x)$. The decoherence rate is given by

$$\lambda = \int d^3x d^3y \frac{D_0}{|x - y|} \left( \sum_{i,j} m_{L,i}(x)m_{L,j}(y) + \sum_{i,j} m_{R,i}(x)m_{R,j}(y) \right). \quad (140)$$

Since the cross terms involving $i, j$ are suppressed by a factor of inter-atomic scales, to leading order the contribution to the decoherence rate is lower bounded by the $i = j$ component of the sums in Eq.

(140), which gives an extra factor of $N$ relative to the single particle case $\lambda = \frac{6D_0NM^2}{5R}$.

Both Eqs. (138) and (139) depend on the probability density of the mass. In particular, taking the probability density to be arbitrarily peaked, one finds that the decoherence rate also diverges. This has to be the case: recall from the section "A Trade-off between decoherence and diffusion" that if one considers a particle in a superposition of two arbitrarily peaked probability densities, then there can be an arbitrarily large response in the Newtonian potential around those points. As a consequence, for such states, the decoherence must occur arbitrarily fast, or there must be an arbitrarily large amount of diffusion to cover up the back-reaction and maintain coherence. For the continuous master equation, such as that of Eq. (101) this diffusion must also occur throughout space, although it can depend on the gravitational degrees of freedom. Since divergent energy production throughout space is clearly unphysical, it must be the case that the decoherence rate must also depend on the expected mass density, as is the case for this example. This argument allows us to rule out continuous master equations that have pure Lindbladian terms that predict decoherence rates which remain finite as the mass density becomes arbitrarily peaked since the coupling constant trade-off will demand that an infinite amount of diffusion is required to cover up the back-reaction and maintain coherence. This is the case for the class of models with CSL-type Lindbladian couplings which are phase space independent, for example in Eq. (106).

### Detecting gravitational diffusion

In this section, we show how the diffusion induced on the Newtonian potential can be measured experimentally. metric leads to observable effects, such as variations in the accelerations involved in torsion experiments and stochastic wave production in cosmology. As shown in the main body of the text, in the non-relativistic limit, $c \to \infty$, the CQ dynamics can be approximated by sourcing the Newtonian potential by a random mass term, and in order to maintain the coherence of any mass is superposition, there must be noise in the Newtonian potential such that we cannot tell which element of the superposition the particle will be in

$$\nabla^2 \Phi = 4\pi G[m(x, t) + u(\Phi, \hat{m})J(x, t)], \quad (141)$$

with

$$\mathbb{E}_m[J(x, t)] = 0, \ \mathbb{E}_m[uJ(x, t)uJ(y, t')] = 2\langle D_2(x, y, \Phi)\rangle\delta(t, t'), \quad (142)$$

where $\langle D_2(x, y, \Phi)\rangle := Tr[D_2^{\mu\nu}(x, y, \Phi_b)L_\mu(x)\rho L_\nu^\dagger(y)]$ and $\rho$ is the quantum state for the decohered mass density. The diffusion coefficient in Eq. (142) is chosen in order for the dynamics to have the same moments as the CQ master equation (4). The solution to Eq. (141), having absorbed $u$ into $J$ is given by

$$\Phi(t, x) = - G \int d^3x' \frac{[m(x', t) - u(\Phi, \hat{m})J(x', t)]}{|x - x'|}, \quad (143)$$

where the statistics of $J$ are described by Eq. (142). A formal treatment of solutions to non-linear stochastic integrals of the form Eq. (141) can be found in ref. 82.

One can also verify this behaviour in specific cases. In the continuous model of the section "Continuous master equation", the noise is taken to be Gaussian, and this, as well as the evolution of the quantum state, is what determines the diffusion in Eq. (143). For the class of discrete models, the higher order moments such as $\mathbb{E}_m[J(x, t)J(y, t')J(z, t'')]$ are suppressed by an order parameter[11,26,37] and whenever this is true we expect we can approximate the dynamics of the Newtonian potential by a Gaussian process. Whether this is the case or not, it is the second-order moment that enters into our

discussion of the variance here. As such, for minimally coupled theories, the Newtonian potential will appear to be sourced by a random mass distribution.

In the discrete case, a precise understanding of the effects of the diffusion beyond the Gaussian approximation involves solving the full classical-quantum dynamics, perhaps using the methods of Oppenheim et al.[26]. In Eq. (141) we are also taking the time scale of the diffusion to be faster than the dynamics of the matter distribution. Likewise for the decoherence—we showed in the "Methods" subsection "Decoherence rates" for continuous models the evolution of the quantum state acts to decohere it into a mass density eigenbasis $m(x)$. One could of course also include the quantum state evolution in a simulation of full CQ dynamics, but this is beyond the scope of the current work.

Note that if the moments of the noise process are Galilean invariant, then the theory given by Eq. (141) is Galilean invariant as we would expect in the Newtonian limit. One can further derive Eq. (141) from manifestly diffeomorphism invariant theories[81], such as those of refs. 39,131.

### Table-top experiments

In this section we estimate the variation in force which would be seen in table-top experiments which bounds the diffusion of classical theories of gravity from above, giving a squeezed bound on $D_2$ due to lower bounds on diffusion arising from coherence experiments. We do this for dynamics in Eq. (141), but the methodology is general and could also be used in a full simulation of CQ dynamics.

The variation in force induced on a composite mass is found via

$$\vec{F}_{\text{tot}} = -\int d^3x\, m(x)\nabla\Phi. \tag{144}$$

Using the solution in Eq. (143), the total force can be written

$$\vec{F}_{\text{tot}} = -G\int d^3x\, d^3x'\, m(x)\frac{(\vec{x}-\vec{x}')}{|x-x'|^3}[m(x',t)-J(x',t)]. \tag{145}$$

In reality, we measure time-averaged force by measuring time-averaged accelerations over the time resolution of the experiment $\Delta T \frac{1}{\Delta T}\int_0^{\Delta T} dt F_{\text{tot}}$. The total variation in the force's time-averaged magnitude. The full covariance matrix for various kernels in the Newtonian limit is given in (J. Oppenheim and A. Russo, manuscript in preparation), $\sigma_F^2 := \vec{F}_{\text{tot}} \cdot \vec{F}_{\text{tot}}$ can be written as

$$\sigma_F^2 = \frac{1}{\Delta T}2G^2\int d^3x\, d^3y\, d^3x'\, d^3y'\, m(x)m(y)\frac{(\vec{x}-\vec{x}')\cdot(\vec{y}-\vec{y}')}{|x-x'|^3|y-y'|^3}\langle D_2(x',y',\Phi)\rangle. \tag{146}$$

We shall use Eq. (146) to provide an upper bound on coupling constants of CQ theories for different choices of kernels $D_2(x',y',\Phi)$. Given a choice of functional form of the kernel, all that remains is the strength of the diffusion coupling, which for the translation invariant kernels we consider here takes the form of a single coupling constant $D_2$. We take $D_2$ to be a dimension-full quantity with units $kg^2\, sm^{-3}$ which characterises the rate of diffusion for the conjugate momenta of the Newtonian potential.

For a composite mass, we can approximate the mass density by summing over $N$ individual atoms of mass density $m_i(x)$, $m(x) = \sum_i m_i(x)$. The total force is the given by $\vec{F}_{\text{tot}} = \sum_i \vec{F}_i$, where $\vec{F}_i$ is the force on each individual atom $\vec{F}_i = -\int_V dx\, m_i(x)\nabla\Phi(x)$, and the total variation of force is then $\sigma_F^2 = \mathbb{E}[\sum_{ij}F_iF_j] - \mathbb{E}[\sum_i F_i]^2$.

In general, the squeeze will depend on the functional choice of $D_2(x,y,\Phi)$ on the Newtonian potential. As mentioned in the main body, in the presence of a large background potential $\Phi_b$, such as that of the Earth's, we will often be able to approximate $D_2(x,y,\Phi) = D_2(x,y,\Phi_b)$.

This is true for the kernels with functional dependence of the form $D_2 \sim \Phi^n$, $D_2 \sim \nabla\Phi$, though the approximation does not hold for all kernels, for example, $D_2 \sim \nabla^2\Phi$ which creates diffusion only where there is the mass density. We hereby shall only consider diffusion kernels $D_2(x,y,\Phi_b)$ where the background potential is dominant, leaving more general considerations for future work.

For local translation invariant dynamics for which the background Newtonian potential is dominant, for example, $D_2 \sim \Phi^n$, we have $\langle D_2(x,y,\Phi_b)\rangle = \langle D_2(\Phi_b)\rangle\delta(x,y)$ and we arrive at the expression for the total variation in time-averaged force

$$\sigma_F^2 = \frac{2G^2}{\Delta T}\sum_{ij}\int d^3x\, d^3y\, d^3x'\, m_i(x)m_j(y)\frac{(\vec{x}-\vec{x}')\cdot(\vec{y}-\vec{x}')}{|x-x'|^3|y-x'|^3}\langle D_2(x',\Phi_b)\rangle. \tag{147}$$

To leading order, the integral in Eq. (147) is dominated by the self variation term where $i=j$, since nuclear scales $10^{-15}$ m dominate over inter-atomic scales $10^{-9}$ m, so that $\mathbb{E}[\sum_{ij}F_iF_j] \sim \sum_i\mathbb{E}[F_i^2]$. Approximating the mass density of the atoms as coming from their nucleus, and taking them to be spheres of constant density $\rho$ with radius $r_N$ and mass $m_N$, we find that the integral in Eq. (147) is approximately

$$\sigma_F^2 \sim \frac{NG^2\rho^2 r_N^2}{\Delta T}\int d^3x'\langle D_2(\Phi_b)\rangle. \tag{148}$$

For the class of continuous dynamics $\langle D_2(\Phi_b)\rangle = D_2(\Phi_b)$, since the diffusion is not associated with any Lindblad operators. If there is noise everywhere throughout space, then the integral in Eq. (148) diverges and gives evidence that continuous CQ theories with noise everywhere should be ruled out.

As such, we expect that continuous CQ theory must contain nonlinear terms proportional to the Newtonian potential appearing in Eq. (141), in which case we can approximate $\int dx' D_2$ by $V_b D_2$ where $V_b$ is the volume of the region over which the background Newtonian potential is significant. In total then, we find for continuous local CQ dynamics

$$\sigma_F^2 \sim \frac{D_2 NG^2\rho^2 r_N^2 V_b}{\Delta T}. \tag{149}$$

From this, we can calculate $D_2$ in terms of the total variance of the acceleration $\sigma_a^2 = \frac{\sigma_F^2}{m_{\text{tot}}^2}$ to get a lower bound

$$D_2 \leq \frac{\sigma_a^2 N r_N^4 \Delta T}{V_b G^2}. \tag{150}$$

Standard Cavendish type classical torsion experiments measure accelerations of the order $10^{-7}$ m s$^{-2}$, and we can take the time over which the acceleration is averaged to be that of minutes $\Delta T \sim 10^2$ s, so a very conservative bound is $\sigma_a \sim 10^{-7}$ m s$^{-2}$, whilst $N$ will be $N \sim 10^{26}$ and $r_N \sim 10^{-15}$ m. We take the background Newtonian potential to be that of the Earth and we (conservatively) take $V_b$ to be $V_b \sim r_E^2 h \sim 10^{15}$ m$^3$ where $r_E$ is the Earth's radius and $h$ is the atmospheric height. We see that this bounds $D_2$ from above by $D_2 \leq 10^{-41}$ kg$^2$ sm$^{-3}$.

On the other hand, $D_2$ is bounded from below from interferometry experiments which bound the decoherence rate. From Eq. (137) and the coupling constant trade-off, for the kernel $D_2(x,y) = D_2\delta(x,y)$ we see (ignoring constant factors) that the decoherence rate is found to be

$$\lambda \sim \frac{N_\lambda M_\lambda^2}{V_\lambda D_2}, \tag{151}$$

where $M_\lambda$ is the mass of a composite particle in the interferometry experiment, which is made up of $N_\lambda$ particles, each with volume $V_\lambda$. This

gives rise to the squeeze

$$\frac{\sigma_a^2 N r_N^4 \Delta T}{V_b G^2} \geq D_2 \geq \frac{N_\lambda M_\lambda^2}{V_\lambda \lambda}. \tag{152}$$

Using the numbers from ref. 59, with $M_\lambda \sim 10^{-24}$ kg, $N_\lambda \sim 10^3$, and $V_\lambda \sim 10^{-15} 10^{-15} 10^{-7}$ m$^3 = 10^{-37}$ m$^3$, $\lambda \sim 10^1$ s$^{-1}$ we find that $D_2 \geq 10^{-9}$ kg$^2$ sm$^{-3}$. This suggests that the $D_2(x,y) = D_2\delta(x,y)$ kernel for classical gravity is already ruled out by experiment.

For the local discrete models, such as that of Eq. (104), the theory is less constrained due to the dependence of the diffusion on the mass density. In this case $\langle D_2(\Phi_b) \rangle = \frac{l_P^3}{m_P} D_2(\Phi_b) m(x)$, where the factors of Planck length and Planck mass are to ensure that $D_2(\Phi_b)$ has the required units. We arrive at the upper bound for $D_2$

$$\frac{\sigma_a^2 N r_N^4 \Delta T m_P}{m_N G^2 l_P^3} \geq D_2. \tag{153}$$

Meanwhile, from Eq. (131), and coupling constant trade-off (30) the decoherence rate for local discreet jumping models goes as $\lambda \sim \frac{M_\lambda m_P}{l_P^3 D_2}$, which gives rise to the lower bound for $D_2$. From this, we arrive at the squeeze

$$\frac{\sigma_a^2 N r_N^4 \Delta T}{m_N G^2} \geq \frac{l_P^3 D_2}{m_P} \geq \frac{M_\lambda}{\lambda}, \tag{154}$$

and plugging in the numbers we find the bound given by Eq. (46) which gives rise to the squeeze for local discrete models $10^{-1}$ kgs $\geq \frac{l_P^3}{m_P} D_2 \geq 10^{-25}$ kgs.

We can also consider other diffusion kernels, for example, that of Eq. (118). In this case, for continuous dynamics, we have that $\langle D_2(x,y) \rangle = -l_P^2 D_2(\Phi_b) \nabla^2 \delta(x,y)$. The Lindbladian kernel saturating the coupling constants trade-off at zeroeth order in $\Phi(x)$, is the Diosi–Penrose kernel $D_0(x,y,\Phi_b) = \frac{D_0(\Phi_b)}{|x-y|}$, as we saw in the section "Examples of Kernels saturating the decoherence diffusion coupling constants trade-off". Approximating the masses as spheres of constant density we find from a substitution of the kernel into Eq. (146) that the variation in time-averaged force is given by

$$\sigma_F^2 \sim \frac{l_P^2 G^2 m_N^2 N D_2}{\Delta T r_N^3}. \tag{155}$$

We therefore find a lower bound for $D_2$ in terms of the variation in acceleration

$$D_2 \leq \frac{\Delta T l_P^2 \sigma_a^2 N r_N^3}{G^2}, \tag{156}$$

which for classical torsion experiments $\sigma_a \sim 10^{-7}$ m s$^{-2}$, $T \sim 10^2$ s, $N \sim 10^{26}$ and $r_N \sim 10^{-15}$ m gives $D_2 l_P^2 \lesssim 10^{-9}$ kg m$^{-1}$. On the other hand, for this kernel the decoherence rate can be calculated via Eq. (139)

$$\lambda \sim \frac{N M_\lambda^2}{l_P^2 D_2 R_\lambda}, \tag{157}$$

which gives the squeeze on $D_2$

$$\frac{\Delta T \sigma_a^2 N r_N^3}{G^2} \geq l_P^2 D_2 \geq \frac{N_\lambda M_\lambda^2}{R_\lambda \lambda}. \tag{158}$$

For the numbers used in the main body of the text[59] $M_\lambda \sim 10^{-24}$ kg, $N_\lambda \sim 10^3$, $R_\lambda \sim V^{1/3} = 10^{-12}$ m, $\lambda \sim 10^1$ s, this yields $D_2 l_P^2 \geq 10^{-35}$ kg m$^{-1}$ and so this model is not ruled out by experiment.

In general then, we expect that by simulating full CQ dynamics satisfying the decoherence diffusion trade-off we will be able to squeeze $D_2$ from above and below. We bound $D_2$ from above by studying the effects of diffusion on gravitational experiments, and we bound $D_2$ from below using the coupling constant trade-off and coherence experiments lower bounding the decoherence rate. As we have seen in this section, it appears that classes of continuous CQ hybrid theories of gravity, including models without spatial correlations, are already experimentally ruled out, whilst others, such as the kernels in subsection "Diffeomorphism invariant kernel" require stronger bounds from both gravitational and coherence experiments. We have been very conservative in our estimates, and so we expect a more thourough analysis will tighten the bounds by orders of magnitude.

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

## Acknowledgements

We would like thank Sougato Bose, Joan Camps, Matt Headrick, Isaac Layton, Juan Maldacena, Andrea Russo, Andy Svesko and Bill Unruh for valuable discussions and Lajos Diósi and Antoine Tilloy for their very helpful comments on an earlier draft of this manuscript. J.O. is supported by an EPSRC Established Career Fellowship, and a Royal Society Wolfson Merit Award, C.S. and Z.W.D. acknowledges financial support from EPSRC. This research was supported by the National Science Foundation under Grant No. NSF PHY11-25915 and by the Simons Foundation *It from Qubit* Network. Research at Perimeter Institute is supported in part by the Government of Canada through the Department of Innovation, Science and Economic Development Canada and by the Province of Ontario through the Ministry of Economic Development, Job Creation and Trade.

## Author contributions

J.O., C.S., B.S. and Z.W.D. contributed to this work.

## Competing interests
