## [Peer Review File · Nature Communications]

Gravitationally induced decoherence vs space-time diffusion: testing the quantum nature of gravityREVIEWER COMMENTS

Reviewer #1 (Remarks to the Author):

The paper discusses the dynamics of composite quantum systems and explores scenarios where one system can be treated classically while the other is quantum-mechanical addressing the classical-quantum (CQ) dynamics of such systems. The authors mention how various attempts to describe CQ dynamics have faced challenges, such as non-positivity and non-linearity. The work focuses on a specific master equation that overcomes these issues and presents consistent dynamics. This master equation is related to the Lindblad equation and is considered the most general Markovian dynamics for an open quantum system with classical-quantum coupling.

The main goal of the work is to explore the common features of CQ dynamics and the experimental signatures that arise from them. It proves that CQ interactions induce decoherence on the quantum system and that there is a trade-off between the rate of decoherence and the amount of diffusion in the classical phase space. This trade-off allows the preservation of quantum coherence despite classical-quantum interactions, evading previous arguments against quantum-classical coupling.

The paper discusses the experimental implications of this trade-off in the context of gravity, and suggests that deviations from Newtonian gravity could be detected by future tabletop experiments. Experimental bounds on the coherence time of large molecules and gravitational experiments measuring the acceleration of small masses already provide significant restrictions on theories where spacetime is treated classically. In particular the bounds obtained by the authors appear to rule out non-relativistic CQ theories of gravity in which the quantum back-reaction is characterized by continuous trajectories in phase space.

The results presented in the work are certainly of much interest as they provide significant insights on the nature of classical-quantum interactions and their experimental consequences concerning the hypothesis of a fundamentally classical nature of gravity. There are however some points which the authors should address in order to clarify some conceptual aspects and consequences of the model of CQ dynamics they study.

The main result which leads to the experimental bounds derived in the work is the equation of motion (39) which, in the non-relativistic limit, reproduces the Poisson equation for the Newtonian potential appended by a diffusion term governed by a stochastic process. This correction term inevitably spoils the covariance of the Poisson equation under transformations belonging to the Galilean group (see e.g. *Class.Quant.Grav.* 28 (2011) 105011, e-Print: 1011.1145 [hep-th]) signalling a breaking of Galilean symmetry in the model. More generally, in the relativistic regime, the same effects should lead to a breaking of Lorentz symmetry. Violations of Lorentz symmetry have been tightly constrained from multiple experimental observations (see e.g. *Living Rev.Rel.* 8 (2005) 5, e-Print: gr-qc/0502097 [gr-qc]) so it is crucial to understand the energy scale at which the breaking of Galilean and Lorentz symmetry occurs in these models and whether existing bounds on Lorentz invariance violation already set stronger restrictions on gravity models with CQ dynamics.

Another aspect deserving attention concerns semiclassical gravity which is briefly mentioned in the Introduction. In its simplest possible form, namely quantum fields on curved space-time, this model has led to the prediction of black hole quantum radiance and it is at the basis of our understanding of the generation of primordial fluctuations in the early universe. One would naturally expect that backreaction of quantum fields on the space-time geometry to be governed by some form of CQ dynamics of the type discussed in the paper. Can the bounds derived in the paper be used to draw conclusions on the type of backreaction to be expected in this setting? For example: would one be able to rule out continuous backreaction on the geometry and thus deduce some discrete dynamics of the geometry already at such semiclassical level?

Concerning the exposition and the way the paper is organized I have the following suggestion for improving the flow and readability of the manuscript:

- It appears that the material in Section II is just a review of known results in the literature while the content of the remaining sections concerns new results. The authors should clearly spell out, especially in Section III and IV, which results are original in their analysis.

- The whole Subsection IIA is devoted to the re-writing of equation (4) in terms of the so-called Kramers-Moyal expansion. I believe it would benefit the readability of the paper if some of the details of such rewriting would be moved to the appendix and the material of this sub-section blended in Subsection IIB.

- In Section III, below equation (18), the authors write the CQ operator $O_{1 \mu}$ in terms of the function $b^i(z-z')$. It appears that the definition of this function is given in Appendix A but it would be helpful for the reader to have it defined also here. The same applies to the field/function A_{μ} appearing in equation (28) whose definition is not clear in the main text.

- I believe it would be beneficial to enhance the clarity of the exposition by having the authors provide illustrative examples of master equations that comply with the trade-off condition specified in equation (23), directly within Section IV, as opposed to relegating them to Appendix D.

Overall the manuscript showcases significant potential and offers valuable insights into the dynamics of composite CQ systems and their experimental consequences when the classical system is the gravitational field. The authors should however thoroughly address my comments and recommendations before the paper is considered for publication in Nature Communications.

Reviewer #2 (Remarks to the Author):

In the present manuscript, the Authors consider a system where one part is treated classically and the other part is treated quantum mechanically. They prove that classical-quantum dynamics necessarily results in decoherence of the quantum system. Furthermore, such effect is accompanied by a breakdown in predictability in the classical phase space. They also prove that there is a trade-off between the rate of decoherence and the degree of diffusion induced in the classical system. This trade-off relation can be used to place experimental constraints on theories where gravity is fundamentally classical. The Authors conclude that the parameter space for such theories is already strongly constrained,

providing some guidance for future investigations.

The paper is well written and the topic is timely and interesting. It is well organized, first presenting a review on how to describe classical-quantum dynamics, while describing the state of the art, and then applying it to the problem at hand. The concluding section summarizes the results, highlighting the main points. Finally, the paper contains all the references necessary for a thorough understanding of the problem.

In my opinion, the paper can be accepted as is.

Reviewer #3 (Remarks to the Author):

The manuscript studies implications arising from a generic quantum system interacting with a classical system which is taken to be general relativity in the weak field limit (Newtonian gravity). As its main result, it argues that torsion balance experiments and coherence times measured in quantum systems actually bound the parameters that couple the two systems. Personally, I find the arguments developed by the authors interesting and convincing. Moreover, the fact that the approach may lead to insights on the quantum nature of spacetime is interesting to a broad audience. As a result, I recommend the publication of the manuscript. Nevertheless, I would like to suggest a small number of improvements which would help in making the manuscript more accessible for a broader audience (see below).

The work is theoretical in nature. There is no data-analysis or experimental setup underlying the published results. All computations are described in sufficient detail so that they can be followed rather easily and checked explicitly. There are only a small number of typos in formulas (see, e.g., eq. (17) or eq. (30)) which do not have any effect on the conclusions. On this basis, I rate the work as technically correct.

Suggested improvements on the manuscript:

1) The theoretical considerations made in particular in section 4 are kept very general. Many of the actual physics applications have been relegated to the appendices. In my opinion, it

would be helpful for the reader, if the authors could add some comments about what the quantities appearing in the abstract formulation will be identified with later on.

2) An interesting feature in eq. (39) is the occurrence of a diffusion process modelled by adding a stochastic random process. In this light, it would be interesting to understand the physical origin of this term. Could it, e.g., be generated by a quantum description of spacetime and what would be the relevant quantity that should be computed in this case?

REVIEWER COMMENTS

Reviewer #1 (Remarks to the Author):

The paper discusses the dynamics of composite quantum systems and explores scenarios where one system can be treated classically while the other is quantum-mechanical addressing the classical-quantum (CQ) dynamics of such systems. The authors mention how various attempts to describe CQ dynamics have faced challenges, such as non-positivity and non-linearity. The work focuses on a specific master equation that overcomes these issues and presents consistent dynamics. This master equation is related to the Lindblad equation and is considered the most general Markovian dynamics for an open quantum system with classical-quantum coupling.

The main goal of the work is to explore the common features of CQ dynamics and the experimental signatures that arise from them. It proves that CQ interactions induce decoherence on the quantum system and that there is a trade-off between the rate of decoherence and the amount of diffusion in the classical phase space. This trade-off allows the preservation of quantum coherence despite classical-quantum interactions, evading previous arguments against quantum-classical coupling.

The paper discusses the experimental implications of this trade-off in the context of gravity, and suggests that deviations from Newtonian gravity could be detected by future tabletop experiments. Experimental bounds on the coherence time of large molecules and gravitational experiments measuring the acceleration of small masses already provide significant restrictions on theories where spacetime is treated classically. In particular the bounds obtained by the authors appear to rule out non-relativistic CQ theories of gravity in which the quantum back-reaction is characterized by continuous trajectories in phase space.

The results presented in the work are certainly of much interest as they provide significant insights on the nature of classical-quantum interactions and their experimental consequences concerning the hypothesis of a fundamentally classical nature of gravity. There are however some points which the authors should address in order to clarify some conceptual aspects and consequences of the model of CQ dynamics they study.

The main result which leads to the experimental bounds derived in the work is the equation of motion (39) which, in the non-relativistic limit, reproduces the Poisson equation for the Newtonian potential appended by a diffusion term governed by a stochastic process. This correction term inevitably spoils the covariance of the Poisson equation under transformations belonging to the Galilean group (see e.g. *Class.Quant.Grav.* 28 (2011) 105011, e-Print: 1011.1145 [hep-th]) signalling a breaking of Galilean symmetry in the model.

More generally, in the relativistic regime, the same effects should lead to a breaking of Lorentz symmetry. Violations of Lorentz symmetry have been tightly constrained from multiple experimental observations (see e.g. Living Rev.Rel. 8 (2005) 5, e-Print: gr-qc/0502097 [gr-qc]) so it is crucial to understand the energy scale at which the breaking of Galilean and Lorentz symmetry occurs in these models and whether existing bounds on Lorentz invariance violation already set stronger restrictions on gravity models with CQ dynamics.

We have shown in [https://link.springer.com/article/10.1007/JHEP08\(2023\)163](https://link.springer.com/article/10.1007/JHEP08(2023)163) that Eq 39 (now equation 40) can be derived from a fully diffeomorphism invariant theory. No breaking of local Lorentz invariance need occur. In the case of Galilean invariance, while it's true that any particular realisation of the noise process $j(x,t)$ in Eq 40 breaks invariance, the moments of the distribution need not, and it is only the moments of $j(x,t)$ which the theory specifies. In the case of Lorentz invariance, we can choose $j(x,t)$ to be a Lorentz scalar, with a mean of $\langle j(x,t) \rangle = 0$ and variance $\langle j(x,t)j(y,s) \rangle = D \delta(x,y) \delta(t,s)$, which is a Lorentz invariant statement.

We now write “Note that if the moments of the noise process are Galilean invariant, then the theory given by Equation \eqref{eq: sourcingbymassApp} is Galilean invariant as we would expect in the Newtonian limit. One can further derive \eqref{eq: sourcingbymassApp} from manifestly diffeomorphism invariant theories[82], such as those of [39,137].”

However, with respect to other modifications to the drift term we now note in the conclusion: “All of these modifications would seem to violate Lorentz invariance in some way, and likely lead to observational consequences\cite{mattingly2005modern}.”

Another aspect deserving attention concerns semiclassical gravity which is briefly mentioned in the Introduction. In its simplest possible form, namely quantum fields on curved space-time, this model has led to the prediction of black hole quantum radiance and it is at the basis of our understanding of the generation of primordial fluctuations in the early universe. One would naturally expect that backreaction of quantum fields on the space-time geometry to be governed by some form of CQ dynamics of the type discussed in the paper. Can the bounds derived in the paper be used to draw conclusions on the type of backreaction to be expect in this setting? For example: would one be able to rule out continuous backreaction on the geometry and thus deduce some discrete dynamics of the geometry already at such semiclassical level?

This is a very interesting question, and one which we have been exploring in detail. We have added a line at the end of the conclusion “The regime in which the classical-quantum theory can be regarded as an effective one, is taken up in \cite{UCLQQtoCQ}, both to address the issue of false decoherence, and also to explore the regime in which the classical-quantum theory may be a useful tool to understand the back-reaction of quantum matter on space-time, such as during black-hole evaporation, and during inflation.” As an aside, the back-reaction can be continuous.

Concerning the exposition and the way the paper is organized I have the following suggestion for improving the flow and readability of the manuscript:

- It appears that the material in Section II is just a review of known results in the literature

while the content of the remaining sections concerns new results. The authors should clearly spell out, especially in Section III and IV, which results are original in their analysis.

In the introduction, we now make it clearer that Section II is review by adding the phrase (as derived in \cite{oppenheim_post-quantum_2018,UCLPawula.}) and made clear that Section III and IV contains new material by adding “Our main result is presented in Sec III. We emphasize this again at the start of Section III “In this section we present our main result....”.

- The whole Subsection IIA is devoted to the re-writing of equation (4) in terms of the so-called Kramers-Moyal expansion. I believe it would benefit the readability of the paper if some of the details of such rewriting would be moved to the appendix and the material of this sub-section blended in Subsection IIB.

Section IIA is very short, only containing three equations. We have to present Equation 9 as it is the central moment expansion which we use throughout the paper, while the definition of the moments, Equation 7, is also clearly required in the main body as it defines the central elements of the main theorem (the drift, diffusion, and decoherence). We fear that moving the remaining quarter page to an appendix would be confusing, and would not remove much from the main paper. We therefore ask for some leeway here to keep this short section in the main body.

- In Section III, below equation (18), the authors write the CQ operator $O_{1 \mu}$ in terms of the function $b^i(z-z')$. It appears that the definition of this function is given in Appendix A but it would be helpful for the reader to have it defined also here. The same applies to the field/function A_{μ} appearing in equation (28) whose definition is not clear in the main text.

Done

- I believe it would be beneficial to enhance the clarity of the exposition by having the authors provide illustrative examples of master equations that comply with the trade-off condition specified in equation (23), directly within Section IV, as opposed to relegating them to Appendix D.

We have put in an example in the main body, Eq 24

Overall the manuscript showcases significant potential and offers valuable insights into the dynamics of composite CQ systems and their experimental consequences when the classical system is the gravitational field. The authors should however thoroughly address my comments and recommendations before the paper is considered for publication in Nature Communications.

Reviewer #2 (Remarks to the Author):

In the present manuscript, the Authors consider a system where one part is treated classically and the other part is treated quantum mechanically. They prove that classical-quantum dynamics necessarily results in decoherence of the quantum system. Furthermore, such effect is accompanied by a breakdown in predictability in the classical phase space. They also prove that there is a trade-off between the rate of decoherence and the degree of diffusion induced in the classical system. This trade-off relation can be used to place experimental constraints on theories where gravity is fundamentally classical. The Authors conclude that the parameter space for such theories is already strongly constrained,

providing some guidance for future investigations.

The paper is well written and the topic is timely and interesting. It is well organized, first presenting a review on how to describe classical-quantum dynamics, while describing the state of the art, and then applying it to the problem at hand. The concluding section summarizes the results, highlighting the main points. Finally, the paper contains all the references necessary for a thorough understanding of the problem.

In my opinion, the paper can be accepted as is.

Reviewer #3 (Remarks to the Author):

The manuscript studies implications arising from a generic quantum system interacting with a classical system which is taken to be general relativity in the weak field limit (Newtonian gravity). As its main result, it argues that torsion balance experiments and coherence times measured in quantum systems actually bound the parameters that couple the two systems. Personally, I find the arguments developed by the authors interesting and convincing. Moreover, the fact that the approach may lead to insights on the quantum nature of spacetime is interesting to a broad audience. As a result, I recommend the publication of the manuscript. Nevertheless, I would like to suggest a small number of improvements which would help in making the manuscript more accessible for a broader audience (see below).

The work is theoretical in nature. There is no data-analysis or experimental setup underlying the published results. All computations are described in sufficient detail so that they can be followed rather easily and checked explicitly. There are only a small number of typos in formulas (see, e.g., eq. (17) or eq. (30)) which do not have any effect on the conclusions. On this basis, I rate the work as technically correct.

We have fixed the typos.

Suggested improvements on the manuscript:

1) The theoretical considerations made in particular in section 4 are kept very general. Many of the actual physics applications have been relegated to the appendices. In my opinion, it would be helpful for the reader, if the authors could add some comments about what the quantities appearing in the abstract formulation will be identified with later on.

We have put in an example, Eq 24.

2) An interesting feature in eq. (39) is the occurrence of a diffusion process modelled by adding a stochastic random process. In this light, it would be interesting to understand the physical origin of this term. Could it, e.g., be generated by a quantum description of spacetime and what would be the relevant quantity that should be computed in this case?

This is an interesting question. Although outside the scope of the present work, we have expanded the discussion at the end of the Conclusion, to address this. In particular, we note that if the diffusion was due to the quantum nature of space-time, then we would expect the amount of diffusion to be minimal as is the case of electromagnetism. On the other hand, if space-time is fundamentally classical, then the diffusion must either be fundamental or not describable within quantum or classical theory.

REVIEWERS' COMMENTS

Reviewer #1 (Remarks to the Author):

The authors have adequately addressed all the concerns and suggestions raised in the initial referee report. Based on the authors' comprehensive responses and revisions, I recommend that the paper be accepted for publication in its current form.